# On the Properties and Estimation of Pointwise Mutual Information Profiles

**Paweł Czyż**[*]                                                           *pawelpiotr.czyz@ai.ethz.ch*
*ETH AI Center and Department of Biosystems Science and Engineering*
*ETH Zurich*
*Zurich, Switzerland*

**Frederic Grabowski**[*]                                              *frederic.grabowski@ippt.pan.pl*
*Institute of Fundamental Technological Research*
*Warsaw, Poland*

**Julia E. Vogt**                                                            *julia.vogt@inf.ethz.ch*
*Department of Computer Science*
*ETH Zurich*
*SIB Swiss Institute for Bioinformatics*
*Zurich, Switzerland*

**Niko Beerenwinkel**[†]                                           *niko.beerenwinkel@bsse.ethz.ch*
*Department of Biosystems Science and Engineering*
*ETH Zurich*
*SIB Swiss Institute for Bioinformatics*
*Basel, Switzerland*

**Alexander Marx**[†]                                               *alexander.marx@tu-dortmund.de*
*Research Center Trustworthy Data Science and Security of the University Alliance Ruhr*
*Department of Statistics*
*TU Dortmund University*
*Dortmund, Germany*

**Reviewed on OpenReview:** *https://openreview.net/forum?id=LdflD41Gn8*

## Abstract

The pointwise mutual information profile, or simply profile, is the distribution of pointwise mutual information for a given pair of random variables. One of its important properties is that its expected value is precisely the mutual information between these random variables. In this paper, we analytically describe the profiles of multivariate normal distributions and show that for an expressive family of distributions, termed *Bend and Mix Models*, the profile can be accurately estimated using Monte Carlo methods. We then show how Bend and Mix Models can be used to study the limitations of existing mutual information estimators, investigate the behavior of neural critics used in variational estimators, and understand the effect of experimental outliers on mutual information estimation. Finally, we show how Bend and Mix Models can be used to obtain model-based Bayesian estimates of mutual information, suitable for problems with available domain expertise in which uncertainty quantification is necessary. The accompanying code is available at `https://github.com/cbg-ethz/bmi`.

---

[*]Equal contribution. [†]Joint supervision.

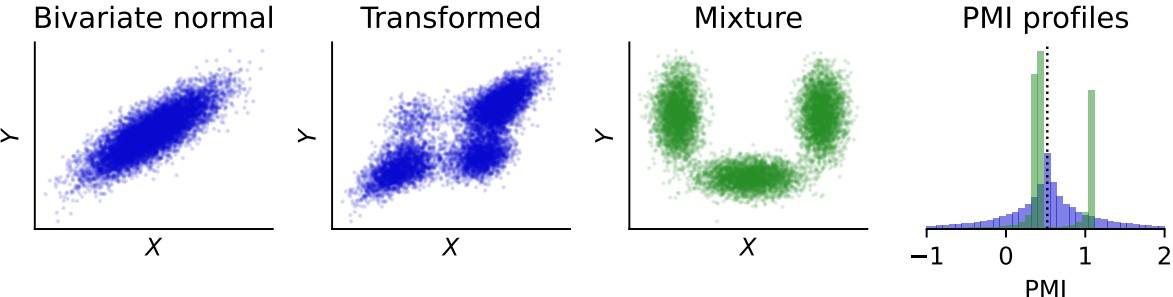

Figure 1: First two panels: samples from a bivariate normal distribution and the same distribution with marginals transformed. Both distributions have the same PMI profile (blue histogram in the fourth panel). Third panel: mixture distribution, which cannot be obtained as a transformation of the normal distribution due to a distinct PMI profile (green histogram in the fourth panel). All three distributions have the same mutual information, marked with the black line in panel four.

## 1 Introduction

Mutual information (MI) is a key measure of dependence between two random variables (r.v.s). Due to its theoretical properties, including invariance to reparametrization, it has found many applications (Polyanskiy & Wu, 2022). However, estimating mutual information from a finite sample remains challenging (McAllester & Stratos, 2020) and the development of mutual information estimators remains an active area of research (Kay, 1992; Kraskov et al., 2004; Oord et al., 2018; Belghazi et al., 2018; Carrara & Ernst, 2023). In response to the introduction of new estimators, various benchmarks have been proposed to assess trade-offs (Khan et al., 2007; Poole et al., 2019; Czyż et al., 2023).

In this manuscript we study the pointwise mutual information profile (PMI profile), a closely related statistical measure which has received much less attention. The PMI profile is a distribution whose expected value is the mutual information between two considered r.v.s. We determine the PMI profile for multivariate normal distributions analytically and show that the PMI profile, similarly to mutual information, is invariant to reparametrization. This last property has an important implication for MI benchmarks (Czyż et al., 2023) which transform simple distributions with known MI to create more complex benchmarking tasks: using the invariance of the profile, we demonstrate that the family of distributions obtained in this manner is inherently limited. For example, distributions such as those studied by Grabowski et al. (2019) and Carrara & Ernst (2023), cannot be obtained by transforming a normal distribution.

To overcome this limitation, we study a family of distributions, here termed Bend and Mix Models (BMMs), for which the PMI profile can be efficiently approximated. By extension, since MI is the expected value of the PMI profile, it can also be readily estimated using Monte Carlo methods. BMMs include highly-expressive Gaussian mixture models, and thus allow us to model complex distributions. Through a series of numerical experiments we demonstrate the usefulness of BMMs for creating non-trivial benchmark tasks; in particular, we discuss robustness of mutual information estimators to inlier and outlier noise. Additionally, BMMs allow us to investigate the properties of PMI profiles directly. Based on empirical evidence, we argue that existing estimators based on neural variational lower bounds (Nguyen et al., 2007; Belghazi et al., 2018) implicitly estimate the PMI profile, even though they do not reliably estimate the pointwise mutual information *function*.

Finally, we show that BMMs can provide model-based Bayesian estimates of both the MI and the PMI profile, generalizing the earlier approaches of Kay (1992) and Brillinger (2004) to a large class of models. Although this approach is not universal, we demonstrate that in low-dimensional problems with available domain expertise, BMMs provide state-of-the-art point estimates. Additionally, they offer a principled approach to uncertainty quantification in MI estimation, which has been lacking in other approaches.

To summarize, the main contributions of this manuscript are as follows:
- We describe the PMI profile, an invariant generalizing mutual information, determine the PMI profile for normal distributions and prove further characterising properties (Sec. 2.1).

- We show that for an expressive family of distributions, called Bend and Mix Models, the PMI profile, and hence mutual information, is tractable using Monte Carlo methods (Sec. 2.2).
- We use Bend and Mix Models to construct novel benchmarking tasks (Sec. 3.1–3.2 and Appendix C.2), study variational MI estimators (Sec. 3.3), and propose model-based Bayesian MI estimators (Sec. 3.4).

## 2 Theoretical framework

In this section, we first introduce PMI profiles and then define Bend and Mix Models, an expressive family of distributions for which the PMI profile can be estimated with Monte Carlo methods.

### 2.1 Pointwise mutual information profiles

Consider random variables $X$ and $Y$ valued in spaces $\mathcal{X}$ and $\mathcal{Y}$. Provided that the joint distribution $P_{XY}$ is sufficiently regular (in the sense of belonging to the family $\mathcal{P}(\mathcal{X}, \mathcal{Y})$ defined in Appendix A.1), the mutual information is given as follows (Pinsker & Feinstein, 1964):

**Definition 1.** *Let $X$ and $Y$ be random variables valued in spaces $\mathcal{X}$ and $\mathcal{Y}$, respectively, such that $P_{XY} \in \mathcal{P}(\mathcal{X}, \mathcal{Y})$. The* mutual information *(MI) is defined as $\mathbf{I}(X; Y) = \mathbb{E}_{(x,y) \sim P_{XY}} [\mathrm{PMI}_{XY}(x, y)]$, where the pointwise mutual information (PMI) function is given as*[1]

$$\mathrm{PMI}_{XY}(x, y) = \log \frac{p_{XY}(x, y)}{p_X(x)\, p_Y(y)},$$

*where $p_{XY}$, $p_X$ and $p_Y$ are PDFs (or PMFs) of the joint and marginal distributions, respectively.*

While mutual information measures whether two variables are independent (which is equivalent to $\mathbf{I}(X; Y) = 0$), the pointwise mutual information function can be understood as a measure of similarity of two events: it is positive if they jointly occur at a higher frequency than if they were sampled independently from the marginal distributions. It is also often associated with variational MI estimators (Nguyen et al., 2007; Belghazi et al., 2018; Oord et al., 2018), where one aims to approximate it using a neural network (cf. Sec. 3.3). However, in this work we focus on the distribution of its values, i.e., its *profile*:

**Definition 2.** *The pointwise mutual information profile*[2] $\mathrm{Prof}_{XY}$ *is defined as the distribution of the random variable $T = \mathrm{PMI}_{XY}(X, Y)$.*

We discuss generalizations of the PMI profile in Appendix A.5. A key property of the PMI profile is that its mean is equal to mutual information, $\mathbf{I}(X; Y) = \mathbb{E}_{T \sim \mathrm{Prof}_{XY}}[T]$. In the following, we characterize the profile for multivariate normal and discrete distributions, and study its invariance properties. It is known that MI is invariant under diffeomorphisms (Kraskov et al., 2004). More generally, in Appendix A.1, we show that the entire profile is invariant:

**Theorem 3.** *Let $P_{XY} \in \mathcal{P}(\mathcal{X}, \mathcal{Y})$ and $f \colon \mathcal{X} \to \mathcal{X}$ and $g \colon \mathcal{Y} \to \mathcal{Y}$ be diffeomorphisms. Then for $X' = f(X)$ and $Y' = g(Y)$ it holds that $P_{X'Y'} \in \mathcal{P}(\mathcal{X}, \mathcal{Y})$ and $\mathrm{Prof}_{XY} = \mathrm{Prof}_{X'Y'}$.*

We demonstrate this result in Fig. 1. The profile of the mixture distribution cannot be obtained as a transformation of the bivariate normal distribution. This, in particular, implies that benchmarks relying on transforming normal distributions for generating more complex problems (Czyż et al., 2023), cannot create distributions with PMI profiles that differ from the normal distributions being transformed.

In fact, for multivariate normal distributions we can express the PMI profile analytically based on the notion of canonical correlations (Hotelling, 1936):

**Theorem 4.** *Let $X$ and $Y$ be r.v.s such that the joint distribution $P_{XY} \in \mathcal{P}(\mathbb{R}^m, \mathbb{R}^n)$ is multivariate normal. If $k = \min(m, n)$ and $\rho_1, \rho_2, \ldots, \rho_k$ are canonical correlations between $X$ and $Y$, then the profile $\mathrm{Prof}_{XY}$ is a*

---

[1] We use the natural logarithm, meaning that all quantities are measured in nats.
[2] Although we are not aware of a prior formal definition and studies of the PMI profile, histograms of approximate PMI between words have been studied before in the computational linguistics community (Allen & Hospedales, 2019).

*generalized $\chi^2$ distribution, namely the distribution of the variable*

$$T = \mathbf{I}(X;Y) + \sum_{i=1}^{k} \frac{\rho_i}{2}(Q_i - Q_i'),$$

*where $\{Q_i, Q_i'\}_{i=1,\ldots,k}$ are independent and identically distributed (i.i.d.) variables drawn from the $\chi_1^2$ distribution. In particular, $\mathrm{Prof}_{XY}$ is symmetric around its median, which coincides with the mean*

$$\mathbf{I}(X;Y) = -\frac{1}{2}\sum_{i=1}^{k}\log\left(1 - \rho_i^2\right),$$

*and all moments of this distribution exist. Its variance is equal to $\mathrm{Var}[T] = \rho_1^2 + \cdots + \rho_k^2$.*

Additionally, in Appendix A.2, we derive bounds on the variance of the PMI profile of multivariate normal distributions given the MI. In the same place, we prove the following results, characterising PMI profiles when MI is zero and for discrete random variables:

**Proposition 5.** *Let $X$ and $Y$ be r.v.s with joint distribution $P_{XY} \in \mathcal{P}(\mathcal{X}, \mathcal{Y})$. Then, $\mathbf{I}(X;Y) = 0$ if and only if $\mathrm{Prof}_{XY} = \delta_0$ is the Dirac measure with a single atom at 0.*

**Proposition 6.** *If $X$ and $Y$ are discrete r.v.s with $P_{XY} \in \mathcal{P}(\mathcal{X}, \mathcal{Y})$, then the PMI profile is discrete:*

$$\mathrm{Prof}_{XY} = \sum_{x \in \mathcal{X}} \sum_{y \in \mathcal{Y}} p_{XY}(x, y)\, \delta_{\mathrm{PMI}_{XY}(x,y)}.$$

For more complex distributions, analytic formulae governing the PMI profiles are not known. Below we address this issue by offering a Monte Carlo approximation for a wide family of distributions.

## 2.2 Bend and Mix Models

As discussed above, PMI profiles are analytically known only for a small number of basic distributions and their reparametrizations, which do not change the PMI profile (Fig. 1). To generate a wider family of distributions with distinct PMI profiles, we introduce *Bend and Mix Models* which combine two strategies: *bending* a distribution (transforming with diffeomorphisms; see also the literature on normalizing flows, e.g., Kobyzev et al. (2021) and Papamakarios et al. (2021)) and *mixing* (combining multiple models into a mixture model). Although the mixing operation generally leads to distributions whose PMI profile is not available analytically, we can efficiently construct numerical approximations via Monte Carlo approaches. To ensure this, we require the following property:

**Definition 7** (informal). *Every distribution $P_{XY} \in \mathcal{P}(\mathcal{X}, \mathcal{Y})$ for which we can efficiently sample $(X, Y) \sim P_{XY}$ and numerically evaluate the densities $p_{XY}(x, y)$, $p_X(x)$ and $p_Y(y)$ at every point $(x, y) \in \mathcal{X} \times \mathcal{Y}$ is considered a Bend and Mix Model (BMM).*

Any distribution that satisfies this definition can be used as a basic building block for a BMM. Examples include discrete distributions as well as multivariate normal and Student distributions. More complex distributions, such as a mixture of discrete and continuous random variables (see discussion and example in Appendix B), can then be constructed using bending and mixing operations:

**Proposition 8.** *If $P_{XY}$ is a BMM and $f$ and $g$ are diffeomorphisms with numerically tractable Jacobians (e.g., normalizing flows), then $P_{f(X)g(Y)}$ is a BMM.*

**Proposition 9.** *Consider BMMs $P_{X_1Y_1}, \ldots, P_{X_KY_K}$. If $w_1, \ldots, w_K$ are positive weights, such that $w_1 + \cdots + w_K = 1$, then the mixture distribution $P_{X'Y'} = w_1 P_{X_1Y_1} + \cdots + w_K P_{X_KY_K}$ is a BMM.*

We prove both propositions in Appendix A.3. Note that since Gaussian mixture models are universal approximators of smooth densities, and they belong to the BMM family, in principle one could use these constructions to approximate any distribution. In practice, the evaluation time of PMI grows linearly with

the number of components and can become costly for a high number of mixture components (or when large normalizing flow architectures are used).

Unfortunately, the MI of a mixture cannot be computed naively by a weighted sum of the information shared within the individual components. To illustrate this, we study two special cases in which constructing a mixture *decreases*, as well as *increases* the overall MI compared to the weighted sum of the components:

**Proposition 10.** *Consider r.v.s $(X_k, Y_k)$ such that $\mathbf{I}(X_k; Y_k) < \infty$ for $k = 1, \ldots, K$. Let $(X', Y')$ be their mixture with weights $w_1, \ldots, w_K$. Then,*

$$0 \leq \mathbf{I}(X'; Y') \leq \sum_{k=1}^{K} w_k \, \mathbf{I}(X_k; Y_k) + \log K.$$

*Moreover, these inequalities are tight:*

1. *There exists a mixture such that $\mathbf{I}(X'; Y') = \log K$ even though $\mathbf{I}(X_k; Y_k) = 0$ for all $k$.*
2. *There exists a mixture such that $\mathbf{I}(X'; Y') = 0$ even though $\mathbf{I}(X_k; Y_k) > 0$ for all $k$.*

More general bounds on MI in mixture models have also been studied by Haussler & Opper (1997) and Kolchinsky & Tracey (2017), but for completeness we include a proof in Appendix A.4.

The properties of BMMs are chosen so that we can estimate the MI using Monte Carlo approaches up to an arbitrary accuracy, rather than relying on upper and lower bounds. We can sample $T \sim \mathrm{Prof}_{XY}$ by sampling a data point $(x, y)$ from the joint distribution $P_{XY}$ and evaluating $t = \mathrm{PMI}_{XY}(x, y)$. Then, MI can be approximated with a Monte Carlo estimate of the integral $\mathbf{I}(X; Y) = \mathbb{E}[T]$. Assuming $\mathbf{I}(X; Y) < \infty$, the Monte Carlo estimator of MI is guaranteed to be unbiased. However, the quality of the estimation depends on the variance of the PMI profile and the number of Monte Carlo samples used. For example, if the variance is infinite, the Monte Carlo estimates (even though unbiased) will not be adequate due to enormous variance and several independent runs may be necessary to diagnose such problems. Additionally, the relative error of the Monte Carlo approximation can be large when the number of samples is not large enough, but the ground-truth MI is close to zero. For a more detailed discussion of Monte Carlo standard error (MCSE) under different regularity conditions we refer to Flegal et al. (2008) and Koehler et al. (2009).

Analogously, we can estimate the PMI profile with a histogram: for a bin $B \subset \mathbb{R}$ one can introduce its indicator function $\mathbf{1}_B$ and integrate $\mathbb{E}[\mathbf{1}_B(T)]$. Its cumulative density function can be approximated with an empirical sample using the expectations $\mathbb{E}[\mathbf{1}_{(-\infty, a_k]}(T)]$ for a given increasing sequence $(a_k)$. Since the indicator functions are bounded between 0 and 1, the Monte Carlo estimator for both quantities is unbiased. For $N$ samples, the standard error is bounded above by $1/\sqrt{4N}$, according to the inequality of Popoviciu (1935).

## 3 Case studies

In this section, we apply Bend and Mix Models to four distinct problems. In Sec. 3.1 we demonstrate how they can be used to extend existing benchmarks of MI estimators (with an explicit benchmark constructed in Appendix C.2). In Sec. 3.2 we show how BMMs can be used to investigate the robustness of MI estimation to outliers and inliers. In Sec. 3.3 we apply BMMs to investigate the biases and sample efficiency of variational estimators employing neural critics. Finally, in Sec. 3.4 we show that if the distribution $P_{XY}$ is known to be well-approximated by a family of BMMs, we can provide Bayesian estimates of both MI and the PMI profile.

### 3.1 Novel distributions for estimator evaluation

To illustrate how BMMs can be used to create expressive benchmark tasks, we implemented a benchmark of 26 continuous distributions in TensorFlow Probability on JAX (Dillon et al., 2017; Bradbury et al., 2018) (Appendix C.2). Additionally, in Appendix B, we consider distributions involving discrete variables. In Fig. 2 we visualise samples from four example distributions: the X distribution is a mixture of two bivariate normal distributions. The marginal distributions $P_X$ and $P_Y$ are normal, although the joint distribution $P_{XY}$ is not. The AI distribution is a mixture of six bivariate normal distributions, illustrating how expressive

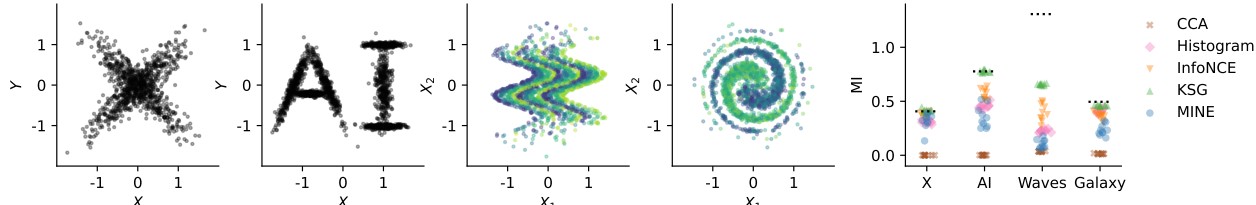

Figure 2: Samples from the example distributions. Distributions X and AI represent one-dimensional variables $X$ and $Y$. Distributions Waves and Galaxy plot two-dimensional $X$ variable using spatial coordinates, while one-dimensional $Y$ variable is represented by color. The rightmost plot presents estimates according to different mutual information algorithms using independently generated data sets with $N = 5\,000$ points each, compared to the ground-truth MI of the distribution (dotted line).

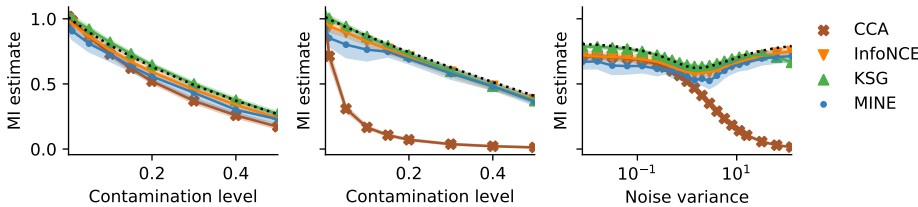

Figure 3: Left: increasing the contamination level $\alpha$ with inlier noise distribution. Middle: increasing the contamination level $\alpha$ with outlier noise distribution. Right: increasing the variance of the noisy normal distribution for constant contamination of 20%. Outliers have less impact than inliers.

BMMs can be. The two final distributions consist of a two-dimensional $X$ variable and a one-dimensional $Y$ variable, allowing us to apply more complex transformations to the $X$ variable. The Waves distribution is a mixture of twelve multivariate normal distributions with the $X$ variable transformed, while the Galaxy distribution is a mixture of two multivariate normal distributions with the $X$ variable transformed by the spiral diffeomorphism (Czyż et al., 2023). A detailed description of these distributions can be found in Appendix C.1.

For each BMM we sampled ten data sets with $N = 5\,000$ points and applied five estimators: the histogram-based estimator (Cellucci et al., 2005; Darbellay & Vajda, 1999), the popular KSG estimator (Kraskov et al., 2004), canonical correlation analysis (CCA; Kay (1992); Brillinger (2004)), and two neural estimators: InfoNCE (Oord et al., 2018) and MINE (Belghazi et al., 2018) (see Appendix C.3 for hyperparameters used). The estimates are shown in Fig. 2.

Even though problems in Fig. 2 are low-dimensional and do not encode more information than 1.5 nats, they pose a considerable challenge for the estimators. The KSG estimator, which performs well in low-dimensional tasks, gave the best estimate in all tasks. However, the Waves task was not solved by any estimator. The CCA estimator, excelling at distributions that are close to multivariate normal (Czyż et al., 2023), is not able to capture any information at all. This suggests that BMMs can provide a rich set of distributions that can be used to test MI estimators. In Appendix C.2 we evaluate KSG, CCA and four neural estimators on the compiled list of 26 proposed benchmark tasks. Particularly interesting are the tasks including inliers, which we study in the next section.

## 3.2 Modeling inliers and outliers

In this section, we use BMMs to study the effect of inliers and outliers on MI estimation. Consider an electric circuit or a biological system modeled as a communication channel $p_{Y|X}(y \mid x)$. The researcher controls the input variable $X$, which results in a distribution $P_X$, and subsequently measures the outcome variable $Y$. The mutual information $\mathbf{I}(X;Y)$ is then estimated from the experimental samples (Nałęcz-Jawecki et al., 2023).

However, every experimental system can suffer from occasional failures. We model the output of a failing system with a noise distribution with a PDF $n(y)$. If the probability of system failure, denoted by $\alpha$, and the distribution of noise $n(y)$ do not depend on the input value $x$, the communication channel becomes a mixture $p_{Y'|X}(y \mid x) = (1 - \alpha)p_{Y|X}(y \mid x) + \alpha n(y)$. By multiplying both sides by the density $p_X(x)$, the distribution of the channel inputs provided by the scientist, we arrive at the mixture distribution $P_{XY'}$ with PDF $p_{XY'}(x, y) = (1 - \alpha)p_{XY}(x, y) + \alpha n(y)p_X(x)$. If the system failure is unnoticed, one can only measure $Y'$, rather than $Y$. It is therefore of interest to understand how much $\mathbf{I}(X; Y)$ and $\mathbf{I}(X; Y')$ can differ under realistic assumptions on the noise $n(y)$ and whether the standard MI estimation techniques are robust to it. In Appendix A.4 we prove the following upper bound:

**Proposition 11.** *Let* $\alpha \in [0, 1]$ *be a parameter. Consider variables* $X$, $Y$ *and* $Y'$, *s.t.* $p_{XY'}(x, y) = (1 - \alpha)p_{XY}(x, y) + \alpha n(y)p_X(x)$. *Then,* $\mathbf{I}(X; Y') \leq (1 - \alpha)\mathbf{I}(X; Y)$.

Alternatively, we can model the system as a BMM and evaluate this quantity using the Monte Carlo approximation. For example, consider a setting with a two-dimensional input variable $X$ and two-dimensional output variables $Y$ (perfect output) and $Y'$ (contaminated output). As the joint density $p_{XY}$ we use a multivariate normal with unit scale and correlations $\mathrm{corr}(X_1, Y_1) = \mathrm{corr}(X_2, Y_2) = 0.8$ and for the noise $n(y)$ we use a multivariate normal distribution with covariance $\sigma^2 I_2$. If $\sigma^2 \approx 1$ this results in inliers, where the noise distribution is hard to distinguish from the signal. For $\sigma^2 \ll 1$ the system failures are all close to 0, while outliers are present for $\sigma^2 \gg 1$.

In Fig. 3 we present the results of three experiments: in the first two, we changed the contamination level $\alpha \in [0, 0.5]$ for $\sigma^2 = 1$ (inlier noise) and $\sigma^2 = 5^2$ (outlier noise) respectively. In the third experiment, we fixed $\alpha = 0.2$ and varied $\sigma^2 \in [2^{-7}, 2^8]$. We see that the inlier noise results in a slightly faster decrease of mutual information, while the outlier noise decreases almost linearly following the upper bound $(1 - \alpha)\mathbf{I}(X; Y)$. Interestingly, in this low-dimensional setting, the KSG, MINE, and InfoNCE estimators reliably estimate the mutual information $\mathbf{I}(X; Y')$, which can significantly differ from $\mathbf{I}(X; Y)$. Although CCA would be the preferred method to estimate $\mathbf{I}(X; Y)$ without any noise in this linear setting (Czyż et al., 2023), even a small number of outliers ($\alpha = 5\%$) can result in unreliable estimates.

### 3.3 Variational estimators and the PMI profile

Variational estimators of mutual information are frequently used in self-supervised learning (Oord et al., 2018) and optimize a critic function $f \colon \mathcal{X} \times \mathcal{Y} \to \mathbb{R}$ to obtain an approximate lower bound on mutual information. For example, Belghazi et al. (2018) use the Donsker–Varadhan loss, $I_{\mathrm{DV}}(f) = \mathbb{E}_{P_{XY}}[f] - \log \mathbb{E}_{P_X \otimes P_Y}[\exp f]$, which is a lower bound on $\mathbf{I}(X; Y)$ for any bounded function $f$. This lower bound becomes tight when $f = \mathrm{PMI}_{XY} + c$ where $c$ is any real number, i.e., $I_{\mathrm{DV}}(f) = \mathbf{I}(X; Y)$. Hence, one can approach MI estimation by optimizing $I_{\mathrm{DV}}$ over a flexible family of functions $f$, usually parameterized by a neural network. Other examples include the NWJ estimator (Nguyen et al., 2007), which becomes tight for $f = \mathrm{PMI}_{XY} + 1$, and InfoNCE (Oord et al., 2018), which has a functional degree of freedom, i.e., $f(x, y) = \mathrm{PMI}_{XY}(x, y) + c(x)$ for a function $c$. We provide more details on these methods in Appendix C.4. Typically, these losses are interpreted as approximate lower bounds on MI and connect $f$ to the PMI function (Poole et al., 2019; Song & Ermon, 2020) in the infinite sample limit. In practice, however, only a finite sample is available and the critic $f$ is modeled via some parametric family (e.g., a neural network), which has to be learned from the data available. Since the ideal critic function of a neural estimator is an approximation of PMI, we investigate how well the PMI function can be learned by the previously described estimators, as well as investigate how they behave under misspecification.

**Critics and the PMI function** We simulated $N = 5\,000$ data points from a mixture of four bivariate normal distributions (Fig. 4) with $\mathbf{I}(X; Y) = 0.36$ and fitted the neural critics (see Appendix C) to half of the data, retaining the latter half as the test set, on which the final estimates were obtained, yielding $I_{\mathrm{NWJ}} = 0.33$, $I_{\mathrm{DV}} = 0.32$, $I_{\mathrm{NCE}} = 0.35$. This (minor) difference can be attributed to the learned critic not matching the true PMI (up to the required constants), estimation error due to evaluation on a finite batch, or both. In Fig. 4 we plot the PMI function and the optimized critics. As the Donsker–Varadhan estimator estimates PMI only up to an additive constant, we normalized the critic and the PMI plots to have zero mean. Analogously, we removed the functional degree of freedom $c(x)$ from the InfoNCE estimator by removing

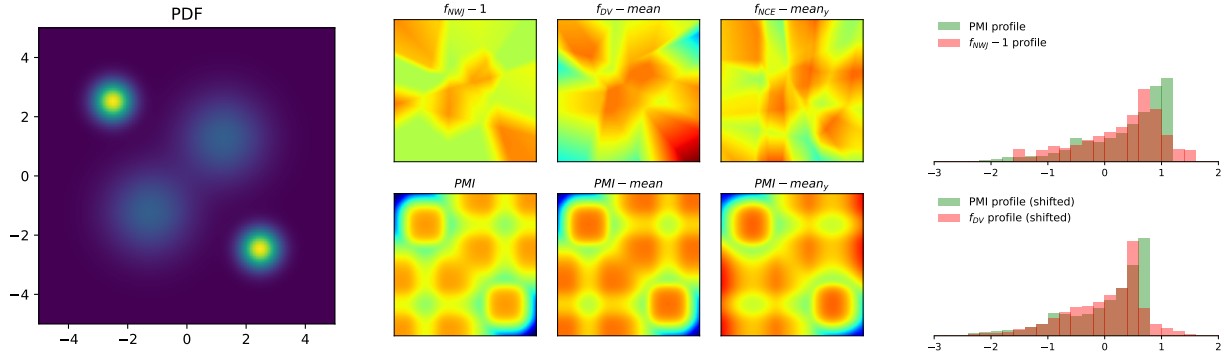

Figure 4: Left: PDF of the considered distribution. Middle: neural critic and PMI values. Right: normalized neural critic and PMI profiles.

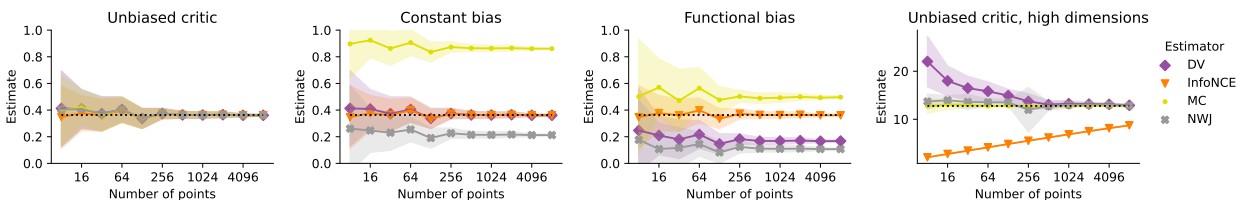

Figure 5: Estimation of mutual information using a function approximating PMI as a function of sample size for Monte Carlo (MC), InfoNCE, Donsker-Varadhan and NWJ losses. From left to right: true PMI function from Fig. 4 is used, a constant bias is added, a functional bias is added. The rightmost plot: true PMI function for a different, high-dimensional problem is used.

the mean calculated along the $y$ dimension. Overall we saw a mismatch, suggesting that neural critics do not capture the PMI function of the distribution well. However, we can compare the PMI profile with the histogram of the values predicted by the critic in Fig. 4 (shifting the PMI profile and the histogram to have mean 0 for the Donsker–Varadhan estimator; for InfoNCE it is not possible to compare the PMI profile with the critic values due to the functional degree of freedom $c(x)$). For both NWJ and DV, we see little discrepancy between the (shifted) PMI profile and learned values. *This suggests that although the neural critics may not learn the PMI function properly in regions with low density, the PMI profile (and, hence, the MI) can still be approximated well.*

**Robustness to misspecification of the critic** In the next experiment (Fig. 5), we evaluated $I_{\mathrm{DV}}$, $I_{\mathrm{NWJ}}$, $I_{\mathrm{InfoNCE}}$, and the simple Monte Carlo estimator (MC) for increasing sample size $N$ where we provided them with an "oracle" critic $f$, which we varied as follows: $f(x,y) - \mathrm{PMI}_{XY}(x,y) \in \{0, c, \sin(x^2)\}$. Using a perfect critic, $f = \mathrm{PMI}_{XY}$, we saw that all estimators performed well in this problem. When a constant bias was added, $f = \mathrm{PMI}_{XY} + c$, the Monte Carlo and NWJ estimators became biased. Finally, InfoNCE was the only unbiased estimator when a functional degree of freedom was added, $f(x,y) = \mathrm{PMI}_{XY}(x,y) + \sin(x^2)$, confirming the known theoretical limitations of the estimators.

**Increasing the Dimensionality** In low dimensions variational approximations perform comparably to the Monte Carlo estimator when the correct critic function is used, and are more robust to misspecification due to additional degrees of freedom. However, this additional robustness results in a bias in higher dimensions (Fig. 5, right panel). We simulated a data set using a multivariate normal distribution with 25 strongly interacting components, $\mathrm{corr}(X_i, Y_i) = 0.8$, which results in $\mathbf{I}(X;Y) \approx 12.8$. Although the Monte Carlo estimator and NWJ estimator (which are not robust to additional degrees of freedom) provided estimates close to the ground-truth (with Monte Carlo slightly outperforming NWJ in both bias and variance), Donsker–Varadhan estimator resulted in a strong positive bias for small batch sizes and InfoNCE had a negative bias with very low variance which has been observed in previous studies and has been related to the $O(\log N)$ behavior of this approximate lower bound (Oord et al., 2018; Song & Ermon, 2020).

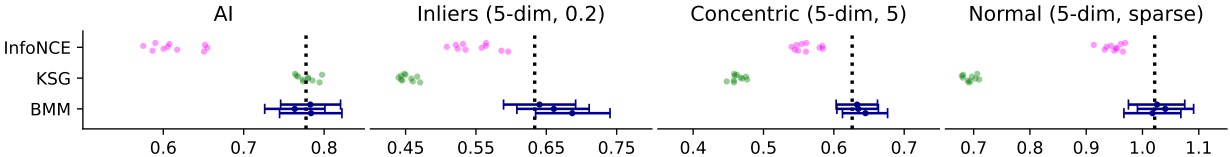

Figure 6: BMMs provide uncertainty estimates in the form of credible intervals and, when domain expertise is available, can be more accurate than generic black-box alternatives. We plot the posterior mean and an interval created using 10th and 90th percentile of the posterior distribution. Ground truth is represented with a dashed line.

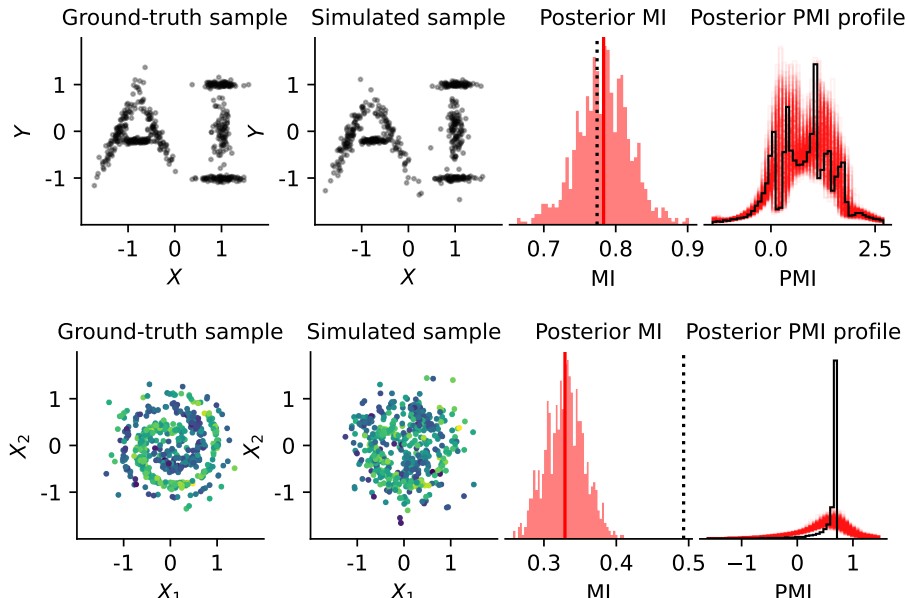

Figure 7: Top row: data generated according to the AI distribution, data generated according to a single MCMC sample, posterior distribution of the mutual information (red line denotes posterior mean and black line denotes the ground-truth value), and the posterior distribution of the PMI profile (black curve denotes the ground-truth value). For a well-specified model the posterior reflects epistemic uncertainty well. Bottom row: experiment with a misspecified model fitted to the Galaxy distribution. Misspecification results in biased inferences (two right-most panels) with miscalibrated posterior, although it can be diagnosed by comparing true data with the data generated from the model.

### 3.4 Model-based mutual information estimation

As the final application of BMMs, we investigate the problem of Bayesian model-based MI estimation.

**The general idea** Consider a family of distributions $\{P_\theta \mid \theta \in \Theta\}$, such that for every value of the parameter vector $\theta$, the mutual information contained in $P_\theta$ is known. That is, for $(X_\theta, Y_\theta) \sim P_\theta$ we can compute $\mathbf{I}(X_\theta; Y_\theta)$, which we denote by $\mathbf{I}(P_\theta)$. If the observed data are modeled as i.i.d. r.v.s $(X_i, Y_i) \sim P_\theta$, one can estimate $\theta$ and use $\mathbf{I}(P_\theta)$ as an estimate of the mutual information. For example, Brillinger (2004) interprets the CCA-based MI estimator of Kay (1992) as the plug-in estimator $\mathbf{I}(P_{\hat{\theta}})$, where $\hat{\theta}$ is the maximum likelihood estimate (MLE) in the model in which $P_\theta$ is a multivariate normal distribution. For discrete data it is well-known the plug-in estimator is biased (Goebel et al., 2005; Suzuki, 2016; Bu et al., 2018; Marx & Vreeken, 2019) and one particular way of regularizing the estimate is to approximate the Bayesian posterior over the MI (Hutter, 2001). We note that one can construct a Bayesian alternative for the CCA-based estimator by using a prior on the covariance matrix (Lewandowski et al., 2009) and Markov chain Monte Carlo (Gelman et al., 2013, Ch. 11) to provide samples $\theta_1, \ldots, \theta_M$ from the posterior $P(\theta \mid X_1, Y_1, \ldots, X_N, Y_N)$ and construct a sample-based approximation to the posterior on mutual information, $\mathbf{I}(P_{\theta_1}), \ldots, \mathbf{I}(P_{\theta_M})$.

More generally, consider a statistical model $\{P_\theta \mid \theta \in \Theta\}$ such that all $P_\theta$ are BMMs, and a prior $P(\theta)$. One can then apply Bayesian inference algorithms to construct a sample $\theta_1, \ldots, \theta_M$ from the posterior. Although the exact values for $\mathbf{I}(P_{\theta_1}), \ldots, \mathbf{I}(P_{\theta_M})$ are not available, they can be approximated as in Sec. 2. Hence, we can construct an approximate posterior distribution and quantify epistemic uncertainty of the estimate (Fig. 7). Since BMMs include both mixture models and normalizing flows, this approach could, in principle, be used as a general technique for building model-based mutual information estimators, where the generative model can be constructed using domain knowledge. Moreover, this is the *first Bayesian estimator of the PMI profile*: as all distributions $P_{\theta_m}$ are BMMs, one can construct $M$ histograms (or CDFs) approximating the profile.

**Proof of concept** To illustrate this approach we implemented a sparse Gaussian mixture model (see Appendix C.5) in NumPyro (Phan et al., 2019) and used the NUTS sampler (Hoffman & Gelman, 2014) to obtain the Bayesian posterior for selected low-dimensional problems from the proposed benchmark (see Table 2 in Appendix C.5.2). We visualise the predictions for four distributions in Fig. 6: we stress that other methods can only provide a single point estimate (for each given data set), while BMMs can provide uncertainty quantification in the form of posterior distribution, visualised by credible intervals. Interestingly, even for low-dimensional problems where KSG has been considered state-of-the-art (Czyż et al., 2023), the BMM-based estimator provided better estimates.

**On the role of assumptions** All parametric statistical methods rely on assumptions (Gelman et al., 2020, Sec. 2). In particular, Bayesian inference can result in unreliable inferences when the model is misspecified, i.e., the true data-generating process $P_{XY}$ does not belong to the assumed family $P_\theta$ (Watson & Holmes, 2016). To illustrate the role of the assumptions, we obtained a Bayesian posterior conditioned on 500 data points from the AI and Galaxy distributions (see Fig. 7). We see that the for the AI distribution, the posterior is concentrated around the ground-truth mutual information value and the ground-truth PMI profile is well-approximated by the posterior samples.

The misspecification in the Galaxy distribution can be immediately diagnosed via posterior predictive checking (Gelman et al., 2013, Ch. 6): in Fig. 7 we see that a data sample simulated from the model looks substantially different from the observed data, meaning that the model did not capture the distribution well (cf. Appendix C.5). This provides a clear indication that the estimates should not be trusted: most of the probability mass of the Bayesian posterior is far from the ground-truth mutual information. Similarly, the posterior on the PMI profile is biased. We therefore recommend using model-checking techniques such as posterior predictive checks and discriminator-based validation (Sankaran & Holmes, 2023, Sec. 4) to understand the deficiencies of the model. To mitigate the risk of overfitting, which can bias the model (see Appendix C.5), we recommend cross-validation (Piironen & Vehtari, 2017).

In summary, although BMMs are not generic estimators applicable to problems without available domain knowledge, in some situations, principled Bayesian modeling can be beneficial to obtain better estimates along with uncertainty quantification.

## 4 Discussion

In this article, we have studied pointwise mutual information profiles (PMI profiles), determining them analytically for multivariate normal distributions (Theorem 4), and proposed the family of Bend and Mix Models (BMMs), which include multivariate normal and Student distributions, mixture models and normalizing flows, for which the PMI profile can be approximated using Monte Carlo methods. We showed how BMMs can be used to provide novel benchmark tasks to test MI estimators and calculate MI transmitted through a communication channel in the presence of inliers and outliers (which can be used in the experimental design in electrical and biological sciences). BMMs allowed us to study how well neural critics approximate PMI profiles. Finally, we showed how Bayesian estimates of MI between continuous r.v.s can be performed using BMMs; additionally, the proposed method estimates the PMI profiles. Although this approach is not universal, we find it suitable for problems with precise domain knowledge available (which can be used to construct the generative distribution $P_\theta$ and provide the prior $P(\theta)$) and in which uncertainty quantification is desired.

**Limitations and further research**   Generalized PMI profiles (see Appendix A.5) allow one to provide a wide family of invariants, which can be related to $f$-divergences (Nowozin et al., 2016). We leave the exploration of potential ramifications of this generalization to future work.

Although in Sec. 3.4 we propose BMMs as a basis for model-based Bayesian inference of mutual information, model misspecification (Watson & Holmes, 2016) as well as underfitting and overfitting can bias inference. We therefore recommend, as usual in Bayesian statistics, to validate the model (Gelman et al., 2020). Overfitting and model misspecification may be more difficult to detect especially in high-dimensional situations or when more expressive models are constructed, for example using normalizing flows, as Bayesian inference for models involving neural networks is known to be challenging (Izmailov et al., 2021). We hypothesise that tempering the likelihood (Grünwald & van Ommen, 2017) or employing alternative inference schemes (Lyddon et al., 2018) may improve robustness to model misspecification, although we leave this topic to future work.

In Sec. 3.2, we study a channel in which the probability of failure $\alpha$ is constant and independent of the input variable $X$. However, in both biological and electric systems this assumption does not necessarily hold: for large values of $X$ the system may result in an outlier more easily. Unfortunately, cases in which $\alpha$ depends on $X$ are not easily modeled by BMMs. Moreover, BMMs do not allow modeling additive noise. Although adding Gaussian noise can be viewed as a continuous mixture, marginalizing the latent variable to obtain an analytic form of $\log p_{Y'}(y)$ is not possible. Employing unbiased estimators, such as SUMO (Luo et al., 2020), may hence become an interesting direction for future research.

More generally, replacing analytical formulae with accurate numerical estimates has the potential to extend the family of distributions with tractable MI beyond BMMs. In Sec. 2.2 we construct BMMs in a bottom-up fashion, where simple distributions are transformed and mixed together to obtain more expressive distributions. The introduced operations allow us to evaluate the probability densities and directly sample from the joint distribution, resulting in a Monte Carlo estimator for mutual information.

Alternatively, one could imagine a top-down approach, where for a given distribution $P_{XY}$ one derives a family of unbiased log-density estimators (Luo et al., 2020). Namely, let $L_{XY}(x,y)$ be a family of estimators indexed by $(x,y) \in \mathcal{X} \times \mathcal{Y}$, such that $\mathbb{E}[L_{XY}(x,y)] = \log p_{XY}(x,y)$ for all points $(x,y)$ in the support of $P_{XY}$. Similarly, let $L_X(x)$ and $L_Y(y)$ be families of estimators such that $\mathbb{E}[L_X(x)] = \log p_X(x)$ and $\mathbb{E}[L_Y(y)] = \log p_Y(y)$. We can then construct an unbiased estimator of the PMI function, $\mathbb{E}[L_{XY}(x,y) - L_X(x) - L_Y(y)] = \mathrm{PMI}_{XY}(x,y)$, and aim to approximate the PMI profile and the ground-truth MI via nested Monte Carlo sampling (Rainforth et al., 2018). However, the feasability of this approach relies on construction of the required families $L_{XY}$, $L_X$ and $L_Y$, as well as on ensuring that the variance of the resulting estimate is sufficiently small.

Similarly, Monte Carlo integration of the PMI function with respect to the $P_{XY}$ measure can, in principle, be replaced by other integration methods, such as Markov chain Monte Carlo (Brooks et al., 2011) or sequential Monte Carlo samplers (Del Moral et al., 2006). These integration methods could allow one to integrate the $\mathrm{PMI}_{XY}$ function for distributions in which the log-density functions $\log p_{XY}$, $\log p_X$ and $\log p_Y$ (or their unbiased estimates) are available, but Monte Carlo sampling from $P_{XY}$ is not possible. However, this approach relies on ensuring whether the employed integration method is accurate enough, e.g., by controlling the convergence diagnostics and the effective sample size (Vehtari et al., 2021; Vats & Knudson, 2021).

We hope that our analysis of pointwise mutual information profiles provides a fresh perspective on mutual information estimators, and that BMMs will prove to be a useful tool for benchmark design and will renew interest in Bayesian approaches to mutual information estimation.

**Reproducibility**   To facilitate reproducibility, the experiments have been implemented as Snakemake workflows (Mölder et al., 2021). The code available at `https://github.com/cbg-ethz/bmi` can be used to reproduce all experimental results and figures.

**Acknowledgments**   FG was supported by the OPUS 18 grant operated by Narodowe Centrum Nauki (National Science Centre, Poland) 2019/35/B/NZ2/03898. PC was supported by a doctoral fellowship from the ETH AI Center. AM was supported by a postdoctoral fellowship from the ETH AI Center and a postdoctoral position at the Computational Biology Group. We would like to thank Paweł Nałęcz-Jawecki, Julia Kostin and anonymous reviewers for helpful suggestions on the manuscript.

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

# Appendix

**Table of Contents**

## A  Technical results

In this section we formalize the results described in Sec. 2. In Sec. A.1 we precisely define the family of $\mathcal{P}(\mathcal{X}, \mathcal{Y})$ distributions, which is then used to prove that the PMI profile is invariant to diffeomorphisms. In Sec. A.2 we derive the PMI profiles in the cases where they are analytically tractable. Then, in Sec. A.3 we formalize the properties of Bend and Mix Models. In Sec. A.4 we derive the bounds on mutual information in finite mixture models. Finally, in Sec. A.5 we provide a measure-theoretic analysis of the PMI profile, which offers more general invariance results, though it requires more complex mathematical machinery.

### A.1  Proof of the invariance of the pointwise mutual information profile

In this section, we consider random variables valued in smooth manifolds without boundary (Lee, 2012, Ch. 1) equipped with their Borel $\sigma$-algebras and given reference measures. In particular, the results in Section 2 apply to both discrete and continuous r.v.s, as the considered manifolds include open subsets of the Euclidean space $\mathbb{R}^n$ with the Lebesgue measure as well as zero-dimensional discrete spaces $\{1, \ldots, m\}$ with the counting measure.

For a given pair of smooth manifolds $\mathcal{X}$ and $\mathcal{Y}$, we equip their product $\mathcal{X} \times \mathcal{Y}$ with the product measure and define the set $\mathcal{P}(\mathcal{X}, \mathcal{Y})$ to consist of all probability measures $P_{XY}$ on $\mathcal{X} \times \mathcal{Y}$ such that the joint measure $P_{XY}$, as well as the marginal measures $P_X(A) = P_{XY}(A \times \mathcal{Y})$ and $P_Y(B) = P_{XY}(\mathcal{X} \times B)$, have smooth and positive PDFs (or PMFs) $p_{XY}$, $p_X$ and $p_Y$ with respect to the reference measures on $\mathcal{X} \times \mathcal{Y}$, $\mathcal{X}$ and $\mathcal{Y}$, respectively. In particular, the PMI function (Definition 1) exists (without the need to define expressions of the form $\log 0/0$) and is a smooth function from the manifold $\mathcal{X} \times \mathcal{Y}$ to the set of real numbers, $\mathbb{R}$. Note that as smooth functions are measurable, the PMI profile, introduced in Definition 2, indeed exists.

More generally, it is possible to work with standard Borel spaces (instead of smooth manifolds) and prove invariance of the profile with respect to arbitrary continuous injective mappings (instead of diffeomorphisms). We discuss this approach in Appendix A.5.

Recall a well-known result (Kraskov et al., 2004, Appendix):

**Lemma 12** (Invariance of PMI). *Let $X' = f(X)$ and $Y' = g(Y)$, where $f$ and $g$ are diffeomorphisms. Then for every $x'$ and $y'$ we have*

$$\mathrm{PMI}_{X'Y'}(x', y') = \mathrm{PMI}_{XY}(x, y),$$

*where $x = f^{-1}(x')$ and $y = g^{-1}(y')$.*

*Proof.* From

$$p_{X'Y'}(x', y') = p_{XY}(x, y) \left| \det D\left(f^{-1} \times g^{-1}\right)(x', y') \right|$$

and analogous quantities we conclude that $p_{X'Y'}$ as well as $p_{X'}$ and $p_{Y'}$ are smooth and everywhere positive functions, so that $\mathrm{PMI}_{X'Y'}$ is well-defined. As $D(f^{-1} \times g^{-1})(x', y')$ is a block matrix with $Df^{-1}(x')$ and $Dg^{-1}(y')$ blocks on the diagonal and other blocks zero, we have $\det D(f^{-1} \times g^{-1})(x', y') = \det Df^{-1}(x') \cdot \det Dg^{-1}(y')$. $\qquad \square$

Now we can prove invariance of the PMI profile:

**Theorem 3.** *Let $P_{XY} \in \mathcal{P}(\mathcal{X}, \mathcal{Y})$ and $f \colon \mathcal{X} \to \mathcal{X}$ and $g \colon \mathcal{Y} \to \mathcal{Y}$ be diffeomorphisms. Then for $X' = f(X)$ and $Y' = g(Y)$ it holds that $P_{X'Y'} \in \mathcal{P}(\mathcal{X}, \mathcal{Y})$ and $\mathrm{Prof}_{XY} = \mathrm{Prof}_{X'Y'}$.*

*Proof.* From the proof of Lemma 12 we conclude that $P_{X'Y'} \in \mathcal{P}(\mathcal{X}, \mathcal{Y})$. Then, we note the profile is the pushforward measure

$$\mathrm{Prof}_{XY} := (\mathrm{PMI}_{XY})_{\#} P_{XY}.$$

Now let $B \subseteq \mathbb{R}$ be any set in the Borel $\sigma$-algebra and $\mathbf{1}_B$ be its characteristic function. Using the change of variables formula for pushforward measure and invariance of PMI:

$$
\begin{aligned}
\mathrm{Prof}_{X'Y'}(B) &= \int \mathbf{1}_B(t)\,\mathrm{d}((\mathrm{PMI}_{X'Y'})_\# P_{X'Y'})(t) \\
&= \int \mathbf{1}_B\left(\mathrm{PMI}_{X'Y'}(x',y')\right)\,\mathrm{d}P_{X'Y'}(x',y') \\
&= \int \mathbf{1}_B\left(\mathrm{PMI}_{X'Y'}(f(x),g(y))\right)\,\mathrm{d}P_{XY}(x,y) \\
&= \int \mathbf{1}_B\left(\mathrm{PMI}_{XY}(x,y)\right)\,\mathrm{d}P_{XY}(x,y) \\
&= \mathrm{Prof}_{XY}(B)
\end{aligned}
$$

$\square$

**Remark 13.** *The above proof can be understood also in the following manner. Lemma 12 allows us to express the PMI function of the transformed variables as the composition $\mathrm{PMI}_{X'Y'} = \mathrm{PMI}_{XY} \circ (f^{-1} \times g^{-1})$. As $\mathrm{Prof}_{X'Y'} = (\mathrm{PMI}_{X'Y'})_\sharp P_{X'Y'}$ and $P_{X'Y'} = (f \times g)_\sharp P_{XY}$, the $f \times g$ cancels out with its inverse:*

$$
\begin{aligned}
\mathrm{Prof}_{X'Y'} &= \left(\mathrm{PMI}_{XY} \circ (f^{-1} \times g^{-1})\right)_\sharp (f \times g)_\sharp P_{XY} \\
&= \left(\mathrm{PMI}_{XY} \circ (f^{-1} \times g^{-1}) \circ (f \times g)\right)_\sharp P_{XY} \\
&= (\mathrm{PMI}_{XY} \circ \mathrm{identity})_\sharp P_{XY} \\
&= (\mathrm{PMI}_{XY})_\sharp P_{XY} = \mathrm{Prof}_{XY}.
\end{aligned}
$$

*This perspective allows one to generalize PMI profile invariance to more general mappings, such as continuous injective functions. We prove a general result in Appendix A.5.*

### A.2 Derivations of pointwise mutual information profiles

The following result shows that the distributions from the class $\mathcal{P}(\mathcal{X}, \mathcal{Y})$ with zero mutual information have the same profile:

**Proposition 5.** *Let $X$ and $Y$ be r.v.s with joint distribution $P_{XY} \in \mathcal{P}(\mathcal{X}, \mathcal{Y})$. Then, $\mathbf{I}(X;Y) = 0$ if and only if $\mathrm{Prof}_{XY} = \delta_0$ is the Dirac measure with a single atom at 0.*

*Proof.* If $\mathrm{Prof}_{XY} = \delta_0$, then the expected value is $\mathbf{I}(X;Y) = 0$. To prove the converse, if $\mathbf{I}(X;Y) = 0$, then $X$ and $Y$ are independent. As the probability density functions of distributions in the $\mathcal{P}(\mathcal{X}, \mathcal{Y})$ class are smooth, we have $p_{XY}(x,y) = p_X(x)p_Y(y)$ at every point $(x,y) \in \mathcal{X} \times \mathcal{Y}$ and $\mathrm{PMI}_{XY}(x,y) = 0$ everywhere. $\square$

The following result characterizes the PMI profiles for discrete r.v.s:

**Proposition 6.** *If $X$ and $Y$ are discrete r.v.s with $P_{XY} \in \mathcal{P}(\mathcal{X}, \mathcal{Y})$, then the PMI profile is discrete:*

$$
\mathrm{Prof}_{XY} = \sum_{x \in \mathcal{X}} \sum_{y \in \mathcal{Y}} p_{XY}(x,y)\,\delta_{\mathrm{PMI}_{XY}(x,y)}.
$$

*Proof.* The measure $P_{XY}$ is discrete and given by

$$
P_{XY} = \sum_{x \in \mathcal{X}} \sum_{y \in \mathcal{Y}} p_{XY}(x,y)\,\delta_{(x,y)},
$$

so its pushforward by the $\mathrm{PMI}_{XY}$ function has the form

$$
\mathrm{Prof}_{XY} = (\mathrm{PMI}_{XY})_\# P_{XY} = \sum_{x \in \mathcal{X}} \sum_{y \in \mathcal{Y}} p_{XY}(x,y)\,\delta_{\mathrm{PMI}_{XY}(x,y)}.
$$

$\square$

The next results describe the properties of PMI profiles associated with multivariate normal variables:

**Theorem 4.** *Let $X$ and $Y$ be r.v.s such that the joint distribution $P_{XY} \in \mathcal{P}(\mathbb{R}^m, \mathbb{R}^n)$ is multivariate normal. If $k = \min(m, n)$ and $\rho_1, \rho_2, \ldots, \rho_k$ are canonical correlations between $X$ and $Y$, then the profile $\mathrm{Prof}_{XY}$ is a generalized $\chi^2$ distribution, namely the distribution of the variable*

$$T = \mathbf{I}(X; Y) + \sum_{i=1}^{k} \frac{\rho_i}{2}(Q_i - Q_i'),$$

*where $\{Q_i, Q_i'\}_{i=1,\ldots,k}$ are independent and identically distributed (i.i.d.) variables drawn from the $\chi_1^2$ distribution. In particular, $\mathrm{Prof}_{XY}$ is symmetric around its median, which coincides with the mean*

$$\mathbf{I}(X; Y) = -\frac{1}{2} \sum_{i=1}^{k} \log\left(1 - \rho_i^2\right),$$

*and all moments of this distribution exist. Its variance is equal to $\mathrm{Var}[T] = \rho_1^2 + \cdots + \rho_k^2$.*

*Proof.* Without loss of generality assume that $m \leq n$. As the PMI profile is invariant to diffeomorphisms (Theorem 3), we can also assume that variables $X$ and $Y$ have been whitened by applying canonical correlation analysis (Jendoubi & Strimmer, 2019), that is $\mathbb{E}[X] = 0$, $\mathbb{E}[Y] = 0$ and the covariance matrix is given by

$$\Sigma = \begin{pmatrix} I_m & \Sigma_{XY} \\ \Sigma_{XY}^T & I_n \end{pmatrix} = \begin{pmatrix} I_m & R & 0 \\ R & I_m & 0 \\ 0 & 0 & I_{n-m} \end{pmatrix}$$

where

$$\Sigma_{XY} = \begin{pmatrix} R & 0_{m \times (n-m)} \end{pmatrix}$$

is an $m \times n$ matrix with the last $n - m$ columns being zero vectors and $R = \mathrm{diag}(\rho_1, \ldots, \rho_m)$ being the $m \times m$ diagonal matrix representing canonical correlations.

We can write the inverse in the block form

$$\Sigma^{-1} = \begin{pmatrix} \Lambda_X & \Lambda_{XY} \\ \Lambda_{XY}^T & \Lambda_Y \end{pmatrix} = \begin{pmatrix} \Lambda_X & \tilde{R} & 0 \\ \tilde{R} & \Lambda_X & 0 \\ 0 & 0 & I_{n-m} \end{pmatrix},$$

where the blocks have been calculated using the formula from Petersen & Pedersen (2012, Sec. 9.1):

$$\Lambda_X = (I_m - \Sigma_{XY}\Sigma_{XY}^T)^{-1} = \mathrm{diag}(u_1, \ldots, u_m)$$
$$\Lambda_Y = (I_n - \Sigma_{XY}^T\Sigma_{XY}) = \mathrm{diag}(u_1, \ldots, u_m, 1, \ldots, 1)$$
$$\Lambda_{XY} = -\Sigma_{XY}\Lambda_Y = \begin{pmatrix} \tilde{R} & 0_{m \times (n-m)} \end{pmatrix},$$

where $\tilde{R} = -\mathrm{diag}(u_1\rho_1, \ldots, u_m\rho_m)$ and $u_i = 1/\left(1 - \rho_i^2\right)$.

We define a quadratic form

$$s(x, y) = x^T \Lambda_X x + y^T \Lambda_Y y + 2x^T \Lambda_{XY} y$$
$$= \sum_{i=1}^{m} u_i\left(x_i^2 + y_i^2 - 2\rho_i x_i y_i\right) + \sum_{j=m+1}^{n} y_j^2$$

which can be used to calculate log-PDFs:

$$\log p_{XY}(x, y) = -\frac{1}{2}s(x, y) - \frac{1}{2}\log\det\Sigma - \frac{m+n}{2}\log 2\pi,$$
$$\log p_X(x) = -\frac{1}{2}x^T x - \frac{m}{2}\log 2\pi,$$
$$\log p_Y(y) = -\frac{1}{2}y^T y - \frac{n}{2}\log 2\pi.$$

Hence,

$$\mathrm{PMI}_{XY}(x,y) = \frac{x^T x + y^T y - s(x,y)}{2} - \frac{1}{2}\log\det\Sigma.$$

We recognize the last summand as

$$\mathbf{I}(X;Y) = \frac{1}{2}\log\left(\frac{\det I_m \cdot \det I_n}{\det \Sigma}\right) = -\frac{1}{2}\log\det\Sigma.$$

Define quadratic form

$$q(x,y) = 2\big(\mathrm{PMI}_{XY}(x,y) - \mathbf{I}(X;Y)\big) = x^T x + y^T y - s(x,y)$$
$$= \sum_{i=1}^{m}\left((1-u_i)\left(x_i^2 + y_i^2\right) + 2\rho_i u_i x_i y_i\right),$$

which has a corresponding matrix

$$Q = \begin{pmatrix} K & F & 0 \\ F & K & 0 \\ 0 & 0 & 0 \end{pmatrix},$$

where

$$K = \mathrm{diag}(1 - u_1, \ldots, 1 - u_m)$$

and

$$F = \mathrm{diag}(\rho_1 u_1, \ldots, \rho_m u_m).$$

We are interested in the distribution of

$$q(X,Y) = \begin{pmatrix} X^T & Y^T \end{pmatrix} Q \begin{pmatrix} X \\ Y \end{pmatrix},$$

where $(X,Y) \sim \mathcal{N}(0,\Sigma)$.

Imhof (1961) presents a general approach to evaluating the distributions of such quadratic forms. Consider a r.v.

$$Z = \begin{pmatrix} \eta \\ \epsilon \\ \xi \end{pmatrix} \sim \mathcal{N}(0, I_{m+n})$$

which is split into blocks of sizes $m$, $m$ and $n - m$. If we construct a linear transformation $A$ such that

$$\begin{pmatrix} X \\ Y \end{pmatrix} = A \begin{pmatrix} \eta \\ \epsilon \\ \xi \end{pmatrix},$$

then the distribution of $q(X,Y)$ is the distribution of

$$Z^T \left(A^T Q A\right) Z, \qquad Z \sim \mathcal{N}(0, I_{m+n}).$$

We can construct $A$ as

$$A = \begin{pmatrix} P_- & P_+ & 0 \\ -P_- & P_+ & 0 \\ 0 & 0 & I_{n-m} \end{pmatrix}$$

where

$$P_- = \mathrm{diag}\left(\sqrt{\frac{1-\rho_1}{2}}, \cdots, \sqrt{\frac{1-\rho_m}{2}}\right), \qquad P_+ = \mathrm{diag}\left(\sqrt{\frac{1+\rho_1}{2}}, \cdots, \sqrt{\frac{1+\rho_m}{2}}\right).$$

We calculate

$$AA^T = \begin{pmatrix} P_-^2 + P_+^2 & P_+^2 - P_-^2 & 0 \\ P_+^2 - P_-^2 & P_-^2 + P_+^2 & 0 \\ 0 & 0 & I_{n-m} \end{pmatrix} = \begin{pmatrix} I_m & R & 0 \\ R & I_m & 0 \\ 0 & 0 & I_{n-m} \end{pmatrix} = \Sigma$$

and

$$A^T Q A = \begin{pmatrix} 2P_-^2(K-F) & 0 & 0 \\ 0 & 2P_+^2(K+F) & 0 \\ 0 & 0 & 0 \end{pmatrix},$$

where

$$2P_-^2(K-F) = \operatorname{diag}\left(-\rho_1, \ldots, -\rho_m\right), \qquad 2P_+^2(K+F) = \operatorname{diag}\left(\rho_1, \ldots, \rho_m\right).$$

Hence, the distribution of $q(X, Y)$ is the same as the distribution of

$$\sum_{i=1}^{m} \rho_i(-\eta_i^2 + \epsilon_i^2) + \sum_{j=1}^{n-m} 0 \cdot \xi_j^2,$$

where $(\eta, \epsilon, \xi) \sim \mathcal{N}(0, I_{m+n})$. To summarize, let $Q_1, \ldots, Q_m, Q_1', \ldots, Q_m'$ be i.i.d. random variables distributed according to the $\chi_1^2$ distribution. The quadratic form $q(X, Y)$ has the same distribution as

$$\sum_{i=1}^{m} \rho_i(Q_i - Q_i'),$$

which can also be written as

$$q(X, Y) \sim \sum_{i=1}^{m} \left(\rho_i \chi_1^2 - \rho_i \chi_1^2\right).$$

Note that this distribution is symmetric around 0. We can now reconstruct the profile from $q(X, Y)$:

$$\operatorname{Prof}_{XY} = \mathbf{I}(X; Y) + \sum_{i=1}^{m} \left(\frac{\rho_i}{2} \chi_1^2 - \frac{\rho_i}{2} \chi_1^2\right),$$

which is symmetric around $\mathbf{I}(X; Y)$ and, in agreement with Proposition 5, degenerates to the atomic distribution $\delta_0$ if and only if $\mathbf{I}(X; Y) = 0$, which is equivalent to $\rho_i = 0$ for all $i$.

As a linear combination of independent $\chi_1^2$ variables, the profile has all finite moments. Using the fact that the variance of $\chi_1^2$ distribution is 2, and quadratic scaling of variance, each term has variance $2 \cdot (\rho_i/2)^2 = \rho_i^2/2$. As variances of independent variables are additive, we can sum up all the $2m$ terms to obtain $\rho_1^2 + \cdots + \rho_m^2$. $\quad\square$

**Proposition 14.** *For a multivariate normal distribution with fixed mutual information, the variance of the PMI profile is maximized when $\rho_1^2 = \rho_2^2 = \ldots = \rho_m^2$.*

*Proof.* Let $a_i = 1 - \rho_i^2$. To maximize the variance we equivalently have to minimize $a_1 + \ldots + a_m$ preserving given constraint on mutual information and $a_i \in (0, 1]$.

The constraint on mutual information takes the form

$$\mathbf{I}(X; Y) = -\frac{1}{2} \sum_{i=1}^{m} \log\left(1 - \rho_i^2\right) = -\frac{1}{2} \log\left(a_1 \cdots a_m\right).$$

Hence, the product $a_1 \cdots a_m$ has to be constant. Denote this constant by $A^m$ for $A \in (0, 1]$ as well. Let $a_1, \ldots, a_m$ be such that $a_1 \cdots a_m = A^m$ and $a_i \in (0, 1]$. From the inequality between arithmetic and geometric means we note that

$$\frac{a_1 + \ldots + a_m}{m} \geq \sqrt[m]{a_1 \cdots a_m} = A,$$

where the equality holds only if $a_1 = \cdots = a_m = A$. Hence, this is the unique minimum under the constraints provided. It follows that $\rho_1^2 = \cdots = \rho_m^2$. $\quad\square$

**Corollary 15.** *For a multivariate normal distribution with fixed mutual information and $m$ canonical correlations, the variance of the PMI profile does not exceed*

$$V = m \left(1 - \exp\left(-2\mathbf{I}(X;Y)/m\right)\right).$$

*Proof.* The variance is maximized when $\rho_1^2 = \cdots = \rho_m^2$. Writing $\rho^2$ for the common value, we have

$$\rho^2 = 1 - \exp\left(-2\mathbf{I}(X;Y)/m\right)$$

and

$$V = m\rho^2 = m \left(1 - \exp\left(-2\mathbf{I}(X;Y)/m\right)\right).$$

The mutual information can also be written as function of variance

$$\mathbf{I}(X;Y) = -\frac{1}{2}m \log\left(1 - \rho^2\right) = -\frac{1}{2}m \log(1 - V/m).$$

$\square$

**Proposition 16.** *For a multivariate normal distribution with fixed non-zero mutual information the variance of the PMI profile is minimized when $\rho_i \neq 0$ for exactly one $i$.*

*Proof.* Let $a_i \in (0,1]$ be any numbers. We have

$$(1 - a_1)(1 - a_2) \geq 0$$

which is equivalent to

$$1 + a_1 a_2 \geq a_1 + a_2$$

where the equality holds if and only if $a_1 = 1$ or $a_2 = 1$.

Using the principle of mathematical induction one can prove a more general inequality:

$$
\begin{aligned}
a_1 a_2 \cdots a_m + (m-1) = 1 + a_1(a_2 \cdots a_m) + (m-2) \\
\geq a_1 + \left(a_2 \cdots a_m + (m-2)\right) \\
\geq a_1 + a_2 + \left(a_3 \cdots a_m + (m-3)\right) \\
\vdots \\
\geq a_1 + a_2 + \cdots + a_m.
\end{aligned}
$$

Let us analyze when the equality can hold. To obtain equality in the first step, we need $a_1 = 1$ or $a_2 \cdots a_m = 1$, which, given the constraints $a_i \in (0,1]$ would mean that $a_2 = \cdots = a_m = 1$. Reasoning inductively, one proves that equality holds only when at least $m-1$ among these numbers are 1.

Now define $a_i = 1 - \rho_i^2$ and note that we are solving a maximization problem $a_i \in (0,1]$ under a constraint

$$a_1 \cdots a_m = P.$$

Using the argument above we note that all the maxima for the above problem are permutations of the sequence $P, 1, 1, \ldots, 1$. This proves that at most one $\rho_i^2 \neq 0$. Hence, we have at most one $\rho_i \neq 0$. $\square$

### A.3 Constructing new Bend and Mix Models

In this section we prove Proposition 8 and Proposition 9.

**Proposition 8.** *If $P_{XY}$ is a BMM and $f$ and $g$ are diffeomorphisms with numerically tractable Jacobians (e.g., normalizing flows), then $P_{f(X)g(Y)}$ is a BMM.*

*Proof.* From the proof of Lemma 12 and the assumption of tractability of the Jacobians of $f$ and $g$ we obtain the tractability of the formulae for the densities $p_{f(X)g(Y)}$, $p_{f(X)}$ and $p_{g(Y)}$. Sampling $(f(X), g(Y))$ amounts to sampling $(X, Y)$ and then transforming the sample using $f$ and $g$. $\qquad\square$

**Proposition 9.** *Consider BMMs $P_{X_1 Y_1}, \ldots, P_{X_K Y_K}$. If $w_1, \ldots, w_K$ are positive weights, such that $w_1 + \cdots + w_K = 1$, then the mixture distribution $P_{X'Y'} = w_1 P_{X_1 Y_1} + \cdots + w_K P_{X_K Y_K}$ is a BMM.*

*Proof.* We can evaluate the densities $p_{X'Y'}(x, y)$, $p_{X'}(x)$ and $p_{Y'}(y)$ of the mixture distribution using the weighted sums of $p_{X_k Y_k}(x, y)$, $p_{X_k}(x)$ and $p_{Y_k}(y)$, respectively. To sample $(X', Y') \sim P_{X'Y'}$ from the mixture distribution we can sample an auxiliary variable $Z \sim \text{Categorical}(K; w_1, \ldots, w_K)$. Then, we have $\big((X', Y') \mid Z = k\big) = (X_k, Y_k)$, so that we can sample from $P_{X_k Y_k}$. $\qquad\square$

### A.4 Mutual information in finite mixtures

In this section we discuss the counterintuitive properties of mutual information in mixture models. We start with the proof of the failing channel inequality:

**Proposition 11.** *Let $\alpha \in [0, 1]$ be a parameter. Consider variables $X$, $Y$ and $Y'$, s.t. $p_{XY'}(x, y) = (1 - \alpha)p_{XY}(x, y) + \alpha n(y)p_X(x)$. Then, $\mathbf{I}(X; Y') \leq (1 - \alpha)\mathbf{I}(X; Y)$.*

*Proof.* Let $Z \sim \text{Bernoulli}(1 - \alpha)$ be an auxiliary variable. We have $(X, Y') \mid (Z = 1) \sim P_{XY}$ and $(X, Y') \mid (Z = 0) \sim P_X \otimes N_Y$.

From the data processing inequality and the chain rule we conclude that

$$\mathbf{I}(X; Y') \leq \mathbf{I}(X; Y', Z) = \mathbf{I}(X; Z) + \mathbf{I}(X; Y' \mid Z).$$

Now note that $X$ and $Z$ are independent, so $\mathbf{I}(X; Z) = 0$.

Hence,

$$
\begin{aligned}
\mathbf{I}(X; Y') &\leq \mathbf{I}(X; Y' \mid Z) \\
&= \alpha \mathbf{D}_{\text{KL}}\left(P_{XY'|Z=0} \parallel P_{X|Z=0} \otimes P_{Y'|Z=0}\right) + (1 - \alpha)\mathbf{D}_{\text{KL}}\left(P_{XY'|Z=1} \parallel P_{X|Z=1} \otimes P_{Y'|Z=1}\right) \\
&= \alpha \mathbf{D}_{\text{KL}}\left(P_{XY'|Z=0} \parallel P_X \otimes N_Y\right) + (1 - \alpha)\mathbf{D}_{\text{KL}}\left(P_{XY} \parallel P_X \otimes P_Y\right) \\
&= \alpha \mathbf{D}_{\text{KL}}\left(P_X \otimes N_Y \parallel P_X \otimes N_Y\right) + (1 - \alpha)\mathbf{I}(X; Y) \\
&= (1 - \alpha)\mathbf{I}(X; Y).
\end{aligned}
$$

$\qquad\square$

In this example, mixing the signal with a noise component, not encoding any information, decreased the information. However, mixing with independent noise components can also *increase* mutual information.

**Example 17.** *Let $A = (0, 1)$ and $B = (1, 2)$ be two disjoint intervals of unit length. We define two pairs of random variables:*

$$(X_1, Y_1) \sim \text{Uniform}(A \times A), \qquad (X_2, Y_2) \sim \text{Uniform}(B \times B).$$

*Note that*

$$\mathbf{I}(X_1; Y_1) = \mathbf{I}(X_2; Y_2) = 0.$$

*If $(X, Y) \sim 0.5 P_{X_1 Y_1} + 0.5 P_{X_2 Y_2}$ is distributed according to a mixture, we have*

$$p_{XY}(x, y) = \frac{1}{2}\mathbf{1}[(x, y) \in A \times A \cup B \times B]$$

*and*

$$p_X(x) = \frac{1}{2}\mathbf{1}[x \in A \cup B], \qquad p_Y(y) = \frac{1}{2}\mathbf{1}[x \in A \cup B].$$

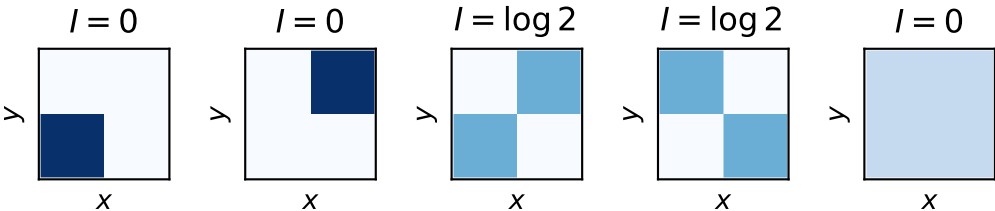

Figure 8: Mixing two distributions with zero MI can result in a mixture distribution with MI equal to one bit. Mixing two one-bit distributions can result in a mixture distribution with no mutual information.

*Hence,*
$$\text{PMI}_{XY}(x, y) = \log 2 \cdot \mathbf{1}[(x, y) \in A \times A \cup B \times B]$$

*and*
$$\mathbf{I}(X; Y) = \frac{1}{2} \log 2 + \frac{1}{2} \log 2 = \log 2.$$

Mixing two distributions with one bit of information each, can however result in independent random variables:

**Example 18.** *Recall the distribution constructed as above:*

$$p_{XY}(x, y) = \frac{1}{2} \mathbf{1}[(x, y) \in A \times A \cup B \times B]$$

*and a symmetric one*

$$p_{UV}(x, y) = \frac{1}{2} \mathbf{1}[(x, y) \in A \times B \cup B \times A].$$

*We have*

$$\mathbf{I}(X; Y) = \mathbf{I}(U; V) = \log 2.$$

*On the other hand, the mixture distribution*

$$(Z, T) \sim 0.5 P_{XY} + 0.5 P_{UV}$$

*has zero mutual information,* $\mathbf{I}(Z; T) = 0$, *as*

$$
\begin{aligned}
p_{ZT}(x, y) &= \frac{1}{4} \mathbf{1}[(x, y) \in (A \cup B) \times (A \cup B)] \\
&= \frac{1}{2} \mathbf{1}[x \in A \cup B] \cdot \frac{1}{2} \mathbf{1}[y \in A \cup B] \\
&= p_Z(x) \cdot p_T(y).
\end{aligned}
$$

Both examples have been visualised in Fig. 8. Note that similarly counterintuitive examples can be constructed using BMMs by using mixtures of multivariate normal distributions.

This demonstrates that mixtures can create and destroy information. There is however an upper bound on the amount of information mixtures can create (see also Haussler & Opper (1997) and Kolchinsky & Tracey (2017)):

**Proposition 10.** *Consider r.v.s* $(X_k, Y_k)$ *such that* $\mathbf{I}(X_k; Y_k) < \infty$ *for* $k = 1, \ldots, K$. *Let* $(X', Y')$ *be their mixture with weights* $w_1, \ldots, w_K$. *Then,*

$$0 \le \mathbf{I}(X'; Y') \le \sum_{k=1}^{K} w_k \mathbf{I}(X_k; Y_k) + \log K.$$

*Moreover, these inequalities are tight:*

1. *There exists a mixture such that* $\mathbf{I}(X'; Y') = \log K$ *even though* $\mathbf{I}(X_k; Y_k) = 0$ *for all* $k$.
2. *There exists a mixture such that* $\mathbf{I}(X'; Y') = 0$ *even though* $\mathbf{I}(X_k; Y_k) > 0$ *for all* $k$.

*Proof.* The examples provide explicit constructions of distributions with the specified properties. To prove the upper bound, consider a variable $Z \sim \text{Categorical}(K; w_1, \ldots, w_K)$. The random variables corresponding to the mixture distribution, $(X', Y')$, have conditional distributions

$$(X', Y') \mid Z = k \sim P_{X_k Y_k}.$$

From the data processing inequality and chain rule we have

$$\mathbf{I}(X'; Y') \leq \mathbf{I}(X'; Y', Z) = \mathbf{I}(X'; Z) + \mathbf{I}(X'; Y' \mid Z).$$

As $Z$ is discrete, the first summand, $\mathbf{I}(X'; Z)$, is bounded from above by the entropy $H(Z)$ (Polyanskiy & Wu, 2022, Th. 3.4e), which cannot exceed $\log K$ (Polyanskiy & Wu, 2022, Th. 1.4b). The second summand can be written as

$$\mathbf{I}(X'; Y' \mid Z) = \sum_{k=1}^{K} P(Z = k) \, \mathbf{D}_{\text{KL}} \left( P_{X'Y'|Z=k} \parallel P_{X'|Z=k} \otimes P_{Y'|Z=k} \right)$$

$$= \sum_{k=1}^{K} w_k \, \mathbf{D}_{\text{KL}} \left( P_{X_k Y_k} \parallel P_{X_k} \otimes P_{Y_k} \right) = \sum_{k=1}^{K} w_k \, \mathbf{I}(X_k; Y_k).$$

$\square$

## A.5  Divergence profiles

In this section we show how to define the PMI profile in a more general setting of a *divergence profile* and prove a general invariance result (Theorem 19). The main ideas stay similar to the proof of Theorem 3.

We consider a standard Borel space $\mathcal{M}$ equipped with two probability measures, $P$ and $Q$, such that $P \ll Q$. For example, if one analyses the mutual information between random variables $X$ and $Y$, they can define $\mathcal{M} = \mathcal{X} \times \mathcal{Y}$ with $P = P_{XY}$ representing the joint distribution and $Q = P_X \otimes P_Y$ representing the product of marginals. Then, a sufficient condition for $P \ll Q$ is to assume that $\mathbf{I}(X; Y) < \infty$.

As $P$ is absolutely continuous with respect to $Q$, the Radon–Nikodym derivative $f = \mathrm{d}P/\mathrm{d}Q$ can be defined. It is a measurable function $f \colon \mathcal{M} \to [0, \infty)$ defined up to a $Q$-null set.

This allows one to define the Kullback–Leibler divergence:

$$\mathbf{D}_{\text{KL}} (P \parallel Q) = \int f \log f \, \mathrm{d}Q = \int \log f \, \mathrm{d}P,$$

where $0 \log 0 = 0$ in the first expression. In the second expression, one can notice that as $P \ll Q$, $\log 0$ can occur only on a $P$-null set. Additionally, using another Radon–Nikodym derivative (differing from $f$ on a $Q$-null set), does not change the value of the integral, so that Kullback–Leibler divergence is indeed well-defined.

These properties correspond to the following properties of the *divergence profile*, defined as the pushforward measure

$$\text{Prof}_{P \parallel Q} = (\log f)_{\sharp} P.$$

Namely, this distribution does not depend on the choice of $f$ (i.e., we can replace $f$ by a function differing on a $Q$-null set) and it is easy to prove that $\text{Prof}_{P \parallel Q}(\{-\infty, \infty\}) = 0$. As this distribution does not have atoms at the infinite values, we can treat it as a distribution over real numbers, $\mathbb{R}$.

Now consider a measurable mapping $i \colon \mathcal{M} \to \mathcal{M}'$ between standard Borel spaces, allowing us to define the pushforward measures $i_{\sharp} P$ and $i_{\sharp} Q$ on $\mathcal{M}'$. We are interested in investigating the profile defined using these pushforward distributions. For example, if $\mathcal{M} = \mathcal{X} \times \mathcal{Y}$, $\mathcal{M}' = \mathcal{X}' \times \mathcal{Y}'$, $P = P_{XY}$, $Q = P_X \otimes P_Y$ and one

considers random variables $X' = i_1(X)$ and $Y' = i_2(Y)$, then the mapping $i = i_1 \times i_2 \colon \mathcal{M} \to \mathcal{M}'$ defines the pushforward measures $P_{X'Y'} = i_\sharp P_{XY}$ and $P_{X'} \otimes P_{Y'} = i_\sharp(P_X \otimes P_Y)$.

Note that the profile $\mathrm{Prof}_{i_\sharp P \| i_\sharp Q}$ can be defined in this case: as $P \ll Q$, then also $i_\sharp P \ll i_\sharp Q$, because $i_\sharp Q(B) = 0$ is equivalent to $Q(i^{-1}(B)) = 0$, which then implies $P(i^{-1}(B)) = 0$.

In general, $\mathrm{Prof}_{i_\sharp P \| i_\sharp Q}$ can differ from $\mathrm{Prof}_{P \| Q}$: for example, using a constant mapping $i$ results in $\mathrm{Prof}_{i_\sharp P \| i_\sharp Q} = \delta_0$, independently on the original profile $\mathrm{Prof}_{P \| Q}$. However, we can generalize Theorem 3 as follows:

**Theorem 19.** *Let $i \colon \mathcal{M} \to \mathcal{M}'$ be a measurable mapping between standard Borel spaces with a measurable left inverse. If $P$ and $Q$ are two probability distributions on $\mathcal{M}$ such that $P \ll Q$, then $\mathrm{Prof}_{i_\sharp P \| i_\sharp Q} = \mathrm{Prof}_{P \| Q}$.*

To prove it, we use the following version of Lemma 12, which generalizes a result of G. (2022):

**Lemma 20.** *Let $i \colon \mathcal{M} \to \mathcal{M}'$ be a measurable mapping between standard Borel spaces such that there exists a measurable left inverse $a \colon \mathcal{M}' \to \mathcal{M}$. If $P$ and $Q$ are two probability distributions on $\mathcal{X}$ such that $P \ll Q$ and $f = \mathrm{d}P/\mathrm{d}Q$ is the Radon–Nikodym derivative, then $i_\sharp P \ll i_\sharp Q$ and the Radon–Nikodym derivative is given by $\mathrm{d}i_\sharp P/\mathrm{d}i_\sharp Q = f \circ a$.*

*Proof of Lemma 20.* Let $a \colon \mathcal{M}' \to \mathcal{M}$ be any measurable left inverse of $i$. To prove that $f \circ a \colon \mathcal{M}' \to [0, \infty)$ is the Radon–Nikodym derivative we need to show that

$$i_\sharp P(B) = \int_B f \circ a \, \mathrm{d}i_\sharp Q$$

for every Borel subset $B \subseteq \mathcal{M}'$. Using the change of variables formula:

$$
\begin{aligned}
\int_B f \circ a \, \mathrm{d}i_\sharp Q &= \int_{i^{-1}(B)} f \circ a \circ i \, \mathrm{d}Q \\
&= \int_{i^{-1}(B)} f \, \mathrm{d}Q \\
&= \int_{i^{-1}(B)} \mathrm{d}P \\
&= P(i^{-1}(B)) = i_\sharp P(B),
\end{aligned}
$$

where the second equality follows from the definition of the left inverse, $a \circ i = \mathrm{id}_\mathcal{M}$.

Note that even if $a$ is substituted for another left inverse, the Radon–Nikodym derivative $f \circ a$ is still determined uniquely up to an $i_\sharp Q$-null set. □

*Proof of Theorem 19.* The profile is defined as $\mathrm{Prof}_{i_\sharp P \| i_\sharp Q} = (\log \mathrm{d}i_\sharp P/\mathrm{d}i_\sharp Q)_\sharp (i_\sharp P)$. Take any measurable left inverse $a$ of $i$ and write

$$\log \frac{\mathrm{d}i_\sharp P}{\mathrm{d}i_\sharp Q} = \log f \circ a,$$

where $f = \mathrm{d}P/\mathrm{d}Q$. We have

$$\mathrm{Prof}_{i_\sharp P \| i_\sharp Q} = (\log f \circ a)_\sharp \, i_\sharp P = (\log f \circ a \circ i)_\sharp P = (\log f)_\sharp P = \mathrm{Prof}_{P \| Q}.$$

□

**Remark 21.** *The above proof suggests that for $P \ll Q$ on a standard Borel space $\mathcal{M}$ and sufficiently well-behaved functions $u \colon [0, \infty) \to \mathbb{R}$, one could define a generalized profile*

$$\left( u \circ \frac{\mathrm{d}P}{\mathrm{d}Q} \right)_\sharp P,$$

*which stays invariant under measurable mappings $i \colon \mathcal{M} \to \mathcal{M}'$ with measurable left inverses.*

*These quantities can be used to study whether a distribution could not be obtained as a simple reparametrization of another one, similarly as the PMI profiles were used in Fig. 1.*

*We suspect that it may also be possible to generalize the profile to use a general $f$-divergence (Polyanskiy & Wu, 2022, Ch. 7), rather than the Kullback–Leibler divergence studied in this manuscript. Nowozin et al. (2016) related $f$-divergences and their variational lower bounds (cf. Sec. 3.3) to generative adversarial networks (GANs), although a relation between a "$f$-divergence profile" and the GAN training remains unclear to us.*

Theorem 19 proves invariance under the existence of a measurable left inverse. Czyż et al. (2023, Theorem 2.1) prove invariance of the mutual information under reparametrizations by continuous injective mappings. As the lemma below shows, Theorem 19 holds also for continuous injective mappings:

**Lemma 22.** *Let $i\colon \mathcal{M} \to \mathcal{M}'$ be a continuous injective mapping between two standard Borel spaces. Then, $i$ admits a measurable left inverse.*

*Proof.* Choose an arbitrary point $x_0 \in \mathcal{M}$ and define a function $a\colon \mathcal{M}' \to \mathcal{M}$ in the following manner:

$$a(y) = \begin{cases} x_0 & \text{if } y \notin i(\mathcal{M}) \\ x & \text{if } x \text{ is the (unique) point such that } i(x) = y \end{cases}$$

This function is well-defined due to the fact that $i$ is injective and it is a left inverse: $a(i(x)) = x$ for all $x \in \mathcal{M}$. Now we need to prove that it is measurable.

Take any Borel set $B \subseteq \mathcal{M}$ and consider its preimage $a^{-1}(B) = \{y \in \mathcal{M}' \mid a(y) \in B\}$. If $x_0 \notin B$, we have $a^{-1}(B) = i(B)$, which is Borel by Lusin–Suslin theorem. If $x_0 \in B$, we can write

$$\begin{aligned} a^{-1}(B) &= a^{-1}(B \setminus \{x_0\}) \cup a^{-1}(\{x_0\}) \\ &= i(B \setminus \{x_0\}) \cup \{i(x_0)\} \cup (\mathcal{M}' \setminus i(\mathcal{M})), \end{aligned}$$

which is Borel as a finite union of Borel sets. $\square$

## B  Distributions involving discrete variables

The formalism in Section 2 is applicable to both continuous and discrete random variables, although in Section 3.1 we focus on the distributions in which both $X$ and $Y$ are continuous. If $X$ and $Y$ are discrete, mutual information $\mathbf{I}(X;Y)$ can be calculated analytically from the joint probability matrix and there exist numerous approaches to estimate it from collected samples (Hutter, 2001; Brillinger, 2004).

In this section we consider the mixed case, in which one variable is continuous and the other one is discrete. For example, Carrara & Ernst (2023) describe a particle physics experiment in which $X$ is an 18-dimensional random variable, but $Y$ is binary. Grabowski et al. (2019) consider a cell transmitting information through the MAPK signalling pathway, assuming the input signal $X$ to be discrete and the measured response $Y$ to be continuous.

### B.1  Known distributions

There are only a few distributions $P_{XY}$ with known ground-truth mutual information assuming this discrete-continuous case. Gao et al. (2017, Sec. 5) describe a discrete random variable $X$ which is uniformly sampled from the set $\mathcal{X} = \{0, \ldots, m-1\}$ and the continuous $Y$ variable is sampled as

$$(Y \mid X = x) \sim \text{Uniform}(x, x+2),$$

which is therefore distributed on $\mathcal{Y} = (0, m+1)$. Gao et al. (2017) prove that mutual information in this case is

$$\mathbf{I}(X;Y) = \log m - \frac{m}{m-1} \log 2$$

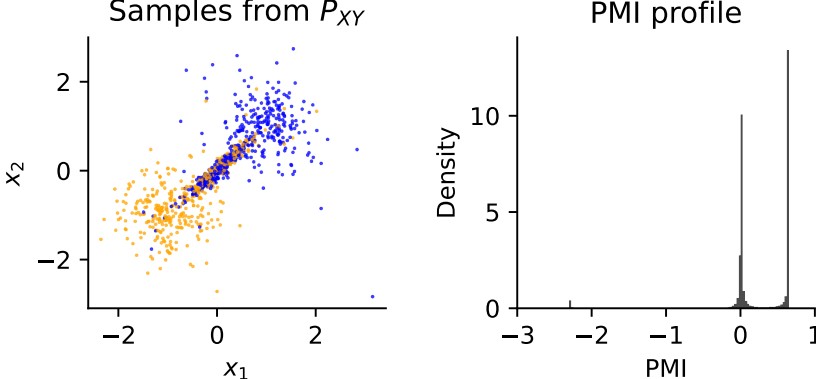

Figure 9: Left: samples from the $P_{XY}$ distribution, colored by the value of the binary variable $Y$. Right: the PMI profile of $P_{XY}$ distribution.

for $m \geq 2$ and $\mathbf{I}(X;Y) = 0$ for $m = 1$ and consider a multivariate analogue of this distribution, in which pairs of variables $(X_k, Y_k)$ for $k = 1, \dots, K$ are sampled independently using the above procedure. Then, they are concatenated into multivariate vectors $(X_1, \dots, X_K)$ and $(Y_1, \dots, Y_K)$ with $K$ times larger mutual information.

We will show how to relate the bivariate example to the framework of Bend and Mix Models (multivariate case can be constructed analogously). Note that the joint distribution $P_{XY}$ is not strictly in $\mathcal{P}(\mathcal{X}, \mathcal{Y})$, as it is supported on the one-dimensional manifold $\mathcal{M} \subset \mathcal{X} \times \mathcal{Y}$ with $m$ connected components (cf. Politis (1991) and Marx et al. (2021)), on which

$$p_{XY}(x, y) = \frac{1}{2m} \mathbf{1}[y \in (x, x+2)],$$

but it is possible to extend the definitions of Section 2, so that this technical difficulty is resolved. The marginal distributions are tractable and admit PDFs:

$$p_X(x) = 1/m,$$
$$p_Y(y) = \sum_{x=0}^{m-1} p_{XY}(x, y).$$

Although $p_Y$ is not smooth on $\mathcal{Y}$ (at integer points), this technical difficulty can also be resolved as this set is of measure zero. Hence, although this distribution is not strictly a BMM, it can still be modeled using the introduced framework.

Next, Gao et al. (2017, Sec. 5) consider the zero-inflated Poissonization of the exponential random variable valued in $\mathcal{X} = \{x \in \mathbb{R} \mid x \geq 0\}$, with: $X \sim \text{Exp}(1)$ and $Y$ being a discrete random variable valued in the set of non-negative integers $\mathcal{Y} = \{0, 1, 2, \dots\}$:

$$(Y \mid X = x) \sim p\, \delta_0 + (1 - p)\, \text{Poisson}(x).$$

They show that

$$\mathbf{I}(X;Y) = (1 - p) \left( 2 \log 2 - \gamma - \sum_{k=0}^{\infty} \log k \cdot 2^{-k} \right),$$

where $\gamma$ is the Euler–Mascheroni constant.

To use BMMs in this case, one has to formally extend the definition of $\mathcal{P}(\mathcal{X}, \mathcal{Y})$ as $\mathcal{X}$ is a manifold with boundary (Lee, 2012, Ch. 1). The joint probability distribution is then given by

$$p_{XY}(x, y) = p_X(x) \cdot p_{Y|X}(y \mid x) = e^{-x} \cdot \left( p \cdot \mathbf{1}[y = 0] + (1 - p) \frac{e^{-x} x^y}{y!} \right)$$

and the PMF function of the $Y$ variable is also analytically known:

$$p_Y(y) = p \cdot \mathbf{1}[y = 0] + (1 - p) \cdot 2^{-(1+y)}$$

Hence, the framework of BMMs, with minor technical adjustments, can accommodate the above distributions.

### B.2 Novel distributions

However, Bend and Mix Models also allow one to create more expressive distributions, for which analytical formulae for ground-truth mutual information are not available, but can be approximated with the Monte Carlo methods as explained in Section 2: consider a continuous random variable $X$ and a discrete random variable $Y$. To introduce a dependency between $X$ and $Y$ variable, we can use a mixture of distributions in which the component variables are independent, i.e., $P_{X_k Y_k} = P_{X_k} \otimes P_{Y_k}$. Therefore, we consider a graphical model $X \leftarrow Z \rightarrow Y$, in which the distributions $P_{X_k} = P_{X|Z=k}$ are known and have tractable PDFs. The distributions of $P_{Y_k} = P_{Y|Z_k}$ are given by probability tables. Monte Carlo estimators of Section 2 can then be used to estimate $\mathbf{I}(X; Y)$ with high accuracy. Note also that this general procedure includes the case $X \leftarrow Y$ by setting $Z = Y$.

We illustrate it in a simple example with $\mathcal{X} = \mathbb{R}^2$, $\mathcal{Y} = \{0, 1\}$ and $K = 3$ components. The first component models a cluster in the $\mathcal{X}$ space, strongly associated with $Y = 1$ value. For $P_{X_1}$ we use a bivariate Student distribution centered at $(1, 1)$ with isotropic dispersion $\Omega = 0.2 \cdot I_2$ and 8 degrees of freedom. We take $P_{Y_1}$ to be the Bernoulli variable with probability $P(Y_1 = 1) = 0.95$.

Analogously, we define a second cluster, strongly associated with $Y = 0$ value: $P_{X_2}$ is a bivariate Student distribution with the same dispersion matrix, but centered at $(-1, -1)$ and with 8 degrees of freedom. Then, $Y_2$ is a Bernoulli variable with $P(Y_2 = 1) = 0.05$.

We then define a third component using a bivariate normal distribution centered at $(0, 0)$ and with covariance matrix

$$\Sigma = 0.1 \begin{pmatrix} 1 & 0.95 \\ 0.95 & 1 \end{pmatrix}.$$

This component is not informative of $Y$, that is $P(Y_3 = 1) = 0.5$.

We used weights $w_1 = w_2 = 1/4$ and $w_3 = 1/2$, what resulted in the distribution visualised in Fig. 9. We estimated both the profile and the mutual information using $N = 10^6$ samples and obtained $\mathbf{I}(X; Y) = 0.224$ with MCSE of $5.1 \cdot 10^{-4}$.

In principle, the above construction can be used to generate realistic high-dimensional data sets (e.g., audio or image) with known ground-truth mutual information, by assuming the generative model $Y \rightarrow X$ (i.e., $Z = Y$) and modeling each $P_{X_k}$ using a normalizing flow or an autoregressive model (Murphy, 2023, Ch. 22) trained on an auxiliary data set with fixed label $Y_k = y_k$. Hence, at least in principle, one could obtain highly-expressive generative process $p_{X|Y}(x \mid y)$ with tractable probability and sampling. Pairing this with an arbitrary probability vector $p_Y(y)$ one can obtain $p_{XY}(x, y)$ and $p_X(x)$ even for high-dimensional data sets, so that Monte Carlo estimator can be used to determine the ground-truth mutual information. However, we anticipate possible practical difficulties with scaling up the proposed approach to high-dimensional data and we leave empirical investigation of this topic to future work.

## C   Experimental protocols and additional experiments

In this section we provide detailed information on experimental protocols used (such as hyperparameter choices) and supplement the findings of Sec. 3 with additional experiments.

### C.1   Specification of the proposed distributions

In this section we provide the details of the distributions introduced in Sec. 3.1. To estimate the ground-truth mutual information we used the Monte Carlo approach described in Sec. 2 with $N = 200\,000$ samples.

### C.1.1  The X distribution

We constructed the X distribution as a mixture of bivariate normal distributions with equal weights, zero mean and covariance matrices specified by

$$\Sigma_\pm = 0.3 \begin{pmatrix} 1 & \pm 0.9 \\ \pm 0.9 & 1 \end{pmatrix}.$$

Note that the marginal distributions of each of component distributions is $\mathcal{N}(0, 0.3^2)$ and subsequently their mixture has exactly the same marginal distributions. This is therefore an interesting example of a distribution in which the joint probability distribution is not multivariate normal, although the marginal distributions of $X$ and $Y$ variables are normal individually. The mutual information is in this case $\mathbf{I}(X; Y) = 0.41$ nats.

### C.1.2  The AI distribution

The AI distribution was constructed as an equally-weighted mixture of six bivariate normal distributions with equal weights and the following parameters:

$$\begin{aligned}
\mu_1 &= (1, 0) \\
\Sigma_1 &= \text{diag}(0.01, 0.2) \\
\mu_2 &= (1, 1) \\
\Sigma_2 &= \text{diag}(0.05, 0.001) \\
\mu_3 &= (1, -1) \\
\Sigma_3 &= \text{diag}(0.05, 0.001) \\
\mu_4 &= (-0.8, -0.2) \\
\Sigma_4 &= \text{diag}(0.03, 0.001) \\
\mu_5 &= (-1.2, 0) \\
\Sigma_5 &= \begin{pmatrix} 0.04 & 0.085 \\ 0.085 & 0.2 \end{pmatrix} \\
\mu_6 &= (-0.4, 0) \\
\Sigma_6 &= \begin{pmatrix} 0.04 & -0.085 \\ -0.085 & 0.2 \end{pmatrix}
\end{aligned}$$

The mutual information of this distribution is $\mathbf{I}(X; Y) = 0.78$ nats.

### C.1.3  The Galaxy distribution

The Galaxy distribution was constructed as an equally-weighted mixture of isotropic multivariate normal distributions with $\mu_\pm = \pm(1, 1, 1)$ and unit covariance matrix and the $X$ variable was transformed using the spiral diffeomorphism with $v = 0.5$, which is described by Czyż et al. (2023). The Galaxy distribution contains $\mathbf{I}(X; Y) = 0.49$ nats.

### C.1.4  The Waves distribution

The Waves distribution was created as an equally-weighted mixture of 12 multivariate normal distributions with equal covariance matrices $\Sigma = \text{diag}(0.1, 1, 0.1)$ and mean vectors

$$\mu_i = (x, 0, x \bmod 4), \quad i \in \{0, 1, \ldots, 11\}.$$

This construction results in a distribution where different vertical components of the $X$ variable are assigned $Y$ values calculated modulo 4. Then, we transformed the $X$ variable with a continuous injection

$$f(x_1, x_2) = (x_1 + 5 \sin(3x_2), x_2),$$

which does not change the mutual information. Finally, we applied the affine mappings

$$a_1(x) = 0.1x - 0.8, \quad a_2(y) = 0.5y,$$

to make the range of the typical values comparable with other distributions. This distribution encodes $\mathbf{I}(X;Y) = 1.31$ nats.

## C.2 A benchmark extension proposal

In Sec. 3.1 and Appendix C.1 we provide examples of novel low-dimensional distributions, which pose a significant challenge to the mutual information estimators. Therefore, we would like to propose a new benchmark of mutual information estimators, extending the benchmark of Czyż et al. (2023).

### C.2.1 Included distributions

The original benchmark of Czyż et al. (2023) consists of 40 distributions. On several problems the estimator performance was similar, so that choosing a representative from a group could allow us to reduce the computation amount. Additionally, we decided to remove the tasks which we considered too difficult: for example, the Student distributions are mapped through the asinh transform to remove the tails. Together with the new tasks, based on BMMs the new version of the benchmark consists of 26 distributions grouped as follows:

**One-dimensional variables** We include four distributions such that $\mathcal{X} = \mathcal{Y} = \mathbb{R}$. From the original benchmark we decided to retain the additive noise task with $\varepsilon = 0.75$ and the asinh-transformed version of the centered bivariate Student distribution with one degree of freedom and the identity matrix for the dispersion (see Czyż et al. (2023, Appendix D) for precise descriptions). Additionally, we include X and AI distributions described in Appendix C.1.

**Embeddings** We retained the Swiss roll embedding distribution (Czyż et al., 2023, Appendix D). In this problem the distribution $P_{XY}$ is supported on a two-dimensional surface embedded in $\mathbb{R}^3$. This distribution does not have a density with respect to the Lebesgue measure on $\mathbb{R}^3$.

**Many-versus-one distributions** We consider multiple distributions with $\mathcal{X} = \mathbb{R}^m$ and $\mathcal{Y} = \mathbb{R}$. For $m = 2$ we consider the Waves and Galaxy distributions described in Appendix C.1. For other $m$ we propose the concentric isotropic multivariate normal distributions. Namely, let $K$ be a parameter, specifying the number of components of a BMM. Each component has independent (multivariate) normal variables $X_k \sim \mathcal{N}(0, k^2 I_m)$ and $Y_k \sim \mathcal{N}(k, 10^{-4})$. To obtain non-zero mutual information we mix these distributions with equal proportions, i.e., the weights vector is given by $w_k = 1/K$ for all $k$. We consider four tasks obtained by varying $m \in \{3, 5\}$ and $K \in \{5, 10\}$. Additionally, we consider a high-dimensional distribution with $m = 25$ and $K = 5$ components.

**Multivariate normal and Student distributions** We selected five problems from multivariate normal distributions (Czyż et al., 2023). We use $\mathcal{X} = \mathbb{R}^m$ and $\mathcal{Y} = \mathbb{R}^n$ and the dense interaction model jointly changing $m = n \in \{5, 25, 50\}$ dimensions and the sparse interactions model (with two pairs of interacting dimensions) in $m = n \in \{5, 25\}$ dimensions.

Additionally, we selected three multivariate Student distributions with the dispersion being the identity matrix. For $m = n = 2$ we consider a distribution with one degree of freedom (i.e., the multivariate Cauchy distributions, which does not have first two moments) and for $m = n \in \{3, 5\}$ we consider a distribution with two degrees of freedom (for which the first moment is defined, but not the second). As described above, the Student distributions had been transformed with the asinh mapping to reduce the tails.

**Spiral diffeomorphism** As described in Czyż et al. (2023), one can transform the multivariate normal distribution with the spiral diffeomorphism, which results in empirically challenging problems. We selected the distributions corresponding to the sparse interactions in $m = n \in \{3, 5\}$ dimensions.

Table 1: Benchmark results. New benchmark problems are marked in green. Best-perfoming estimator in each row has been marked with bold font.

| | CCA | DV | InfoNCE | KSG | MINE | NWJ | True MI |
|---|---|---|---|---|---|---|---|
| AI | $0.00 \pm 0.01$ | $0.57 \pm 0.06$ | $0.61 \pm 0.03$ | $\mathbf{0.78 \pm 0.02}$ | $0.49 \pm 0.07$ | $0.53 \pm 0.10$ | $\mathbf{0.78}$ |
| X | $0.00 \pm 0.01$ | $0.37 \pm 0.02$ | $0.38 \pm 0.02$ | $\mathbf{0.42 \pm 0.02}$ | $0.33 \pm 0.04$ | $0.36 \pm 0.02$ | $\mathbf{0.41}$ |
| Additive | $0.18 \pm 0.01$ | $0.31 \pm 0.02$ | $0.30 \pm 0.02$ | $\mathbf{0.32 \pm 0.01}$ | $\mathbf{0.32 \pm 0.02}$ | $\mathbf{0.32 \pm 0.01}$ | $\mathbf{0.33}$ |
| Swiss roll | $0.02 \pm 0.01$ | $0.37 \pm 0.02$ | $0.37 \pm 0.03$ | $\mathbf{0.42 \pm 0.02}$ | $0.37 \pm 0.03$ | $0.37 \pm 0.02$ | $\mathbf{0.41}$ |
| Galaxy | $0.01 \pm 0.01$ | $0.31 \pm 0.04$ | $0.36 \pm 0.03$ | $\mathbf{0.46 \pm 0.01}$ | $0.26 \pm 0.06$ | $0.23 \pm 0.04$ | $\mathbf{0.49}$ |
| Waves | $0.04 \pm 0.01$ | $0.14 \pm 0.07$ | $0.34 \pm 0.13$ | $\mathbf{0.66 \pm 0.01}$ | $0.13 \pm 0.10$ | $0.11 \pm 0.07$ | $\mathbf{1.31}$ |
| Concentric (3-dim, 10) | $0.00 \pm 0.01$ | $0.50 \pm 0.02$ | $0.50 \pm 0.02$ | $\mathbf{0.51 \pm 0.01}$ | $0.45 \pm 0.02$ | $0.44 \pm 0.03$ | $\mathbf{0.56}$ |
| Concentric (3-dim, 5) | $0.00 \pm 0.01$ | $0.41 \pm 0.03$ | $0.41 \pm 0.02$ | $\mathbf{0.42 \pm 0.02}$ | $0.38 \pm 0.04$ | $0.36 \pm 0.03$ | $\mathbf{0.46}$ |
| Concentric (5-dim, 10) | $0.00 \pm 0.01$ | $0.65 \pm 0.03$ | $\mathbf{0.66 \pm 0.03}$ | $0.55 \pm 0.01$ | $0.62 \pm 0.04$ | $0.56 \pm 0.02$ | $\mathbf{0.75}$ |
| Concentric (5-dim, 5) | $0.00 \pm 0.01$ | $0.55 \pm 0.03$ | $\mathbf{0.56 \pm 0.02}$ | $0.46 \pm 0.01$ | $0.53 \pm 0.03$ | $0.42 \pm 0.05$ | $\mathbf{0.63}$ |
| Concentric (25-dim, 5) | $0.00 \pm 0.01$ | $0.67 \pm 0.06$ | $\mathbf{0.68 \pm 0.04}$ | $0.12 \pm 0.01$ | $0.65 \pm 0.04$ | $0.28 \pm 0.06$ | $\mathbf{1.19}$ |
| Student (1-dim) | $0.00 \pm 0.01$ | $0.17 \pm 0.07$ | $0.20 \pm 0.02$ | $\mathbf{0.22 \pm 0.01}$ | $0.18 \pm 0.01$ | $0.18 \pm 0.06$ | $\mathbf{0.22}$ |
| Student (2-dim) | $0.00 \pm 0.01$ | $0.36 \pm 0.03$ | $\mathbf{0.37 \pm 0.03}$ | $0.36 \pm 0.02$ | $0.27 \pm 0.10$ | $0.24 \pm 0.08$ | $\mathbf{0.43}$ |
| Student (3-dim) | $0.00 \pm 0.01$ | $0.16 \pm 0.08$ | $\mathbf{0.20 \pm 0.03}$ | $0.17 \pm 0.01$ | $0.09 \pm 0.08$ | $0.03 \pm 0.01$ | $\mathbf{0.29}$ |
| Student (5-dim) | $0.01 \pm 0.01$ | $0.21 \pm 0.07$ | $\mathbf{0.28 \pm 0.03}$ | $0.23 \pm 0.01$ | $0.13 \pm 0.12$ | $0.03 \pm 0.03$ | $\mathbf{0.45}$ |
| Inliers (25-dim, 0.2) | $\mathbf{0.58 \pm 0.03}$ | $0.34 \pm 0.04$ | $0.27 \pm 0.09$ | $0.12 \pm 0.03$ | $0.39 \pm 0.05$ | $0.41 \pm 0.04$ | $\mathbf{0.63}$ |
| Inliers (25-dim, 0.5) | $\mathbf{0.24 \pm 0.02}$ | $-0.11 \pm 0.09$ | $-0.08 \pm 0.04$ | $0.05 \pm 0.02$ | $0.08 \pm 0.02$ | $0.06 \pm 0.04$ | $\mathbf{0.27}$ |
| Inliers (5-dim, 0.2) | $0.52 \pm 0.02$ | $0.55 \pm 0.03$ | $0.55 \pm 0.03$ | $0.45 \pm 0.02$ | $0.54 \pm 0.05$ | $\mathbf{0.56 \pm 0.08}$ | $\mathbf{0.63}$ |
| Inliers (5-dim, 0.5) | $0.18 \pm 0.01$ | $0.19 \pm 0.02$ | $0.18 \pm 0.03$ | $0.19 \pm 0.01$ | $0.20 \pm 0.04$ | $\mathbf{0.21 \pm 0.04}$ | $\mathbf{0.27}$ |
| Normal (25-dim, dense) | $\mathbf{1.35 \pm 0.02}$ | $1.09 \pm 0.07$ | $1.06 \pm 0.06$ | $1.05 \pm 0.02$ | $1.16 \pm 0.05$ | $0.18 \pm 0.38$ | $\mathbf{1.29}$ |
| Normal (5-dim, dense) | $\mathbf{0.60 \pm 0.02}$ | $0.55 \pm 0.03$ | $0.54 \pm 0.03$ | $0.56 \pm 0.01$ | $0.56 \pm 0.03$ | $0.56 \pm 0.02$ | $\mathbf{0.59}$ |
| Normal (50-dim, dense) | $1.87 \pm 0.02$ | $1.26 \pm 0.07$ | $1.20 \pm 0.09$ | $1.25 \pm 0.02$ | $\mathbf{1.45 \pm 0.05}$ | $0.62 \pm 0.78$ | $\mathbf{1.62}$ |
| Normal (25-dim, sparse) | $\mathbf{1.08 \pm 0.02}$ | $0.69 \pm 0.05$ | $0.67 \pm 0.07$ | $0.18 \pm 0.02$ | $0.78 \pm 0.05$ | $0.79 \pm 0.03$ | $\mathbf{1.02}$ |
| Normal (5-dim, sparse) | $\mathbf{1.03 \pm 0.02}$ | $0.95 \pm 0.02$ | $0.95 \pm 0.02$ | $0.69 \pm 0.02$ | $0.92 \pm 0.05$ | $0.95 \pm 0.02$ | $\mathbf{1.02}$ |
| Spiral (3-dim) | $0.24 \pm 0.02$ | $0.50 \pm 0.03$ | $0.53 \pm 0.04$ | $\mathbf{0.65 \pm 0.02}$ | $0.43 \pm 0.05$ | $0.45 \pm 0.03$ | $\mathbf{1.02}$ |
| Spiral (5-dim) | $0.39 \pm 0.02$ | $0.48 \pm 0.03$ | $\mathbf{0.52 \pm 0.03}$ | $0.50 \pm 0.01$ | $0.45 \pm 0.03$ | $0.48 \pm 0.03$ | $\mathbf{1.02}$ |

**Inliers** Finally, we implemented four distributions based along the principles described in Sec. 3.2. Namely, we took the multivariate normal distributions with sparse interactions (described above) with $m = n \in \{5, 25\}$ dimensions $P_{XY}$ and then constructed a mixture $(1 - \alpha)P_{XY} + \alpha P_X \otimes P_Y$ for the inlier fraction $\alpha \in \{0.2, 0.5\}$.

### C.2.2    Experimental protocol

**Determining the ground-truth value**   The ground-truth mutual information for distributions proposed by Czyż et al. (2023) is analytically available. For the tasks based on BMMs we used $N = 200,000$ Monte Carlo samples to provide an estimate, as described in Section 2.2. The additional computational cost of estimating the ground-truth mutual information for these distributions is minor: on a standard laptop, estimating the mutual information of the Galaxy task to be $0.4953 \pm 0.0012$ (which results in the relative MCSE smaller than 0.5%, using 200,000 samples) takes less than 2.5 seconds. Similarly, estimating the mutual information of the Concentric task (25-dimensional variant with 5 components) to be $1.1911 \pm 0.0015$ (using 200,000 samples again) takes less than 3.5 seconds.

**Included estimators**   We included a total of six estimators in this benchmark: four variational estimators, the neigbhorhood-based KSG estimator (Kraskov et al., 2004), and the simple CCA-based estimator of Kay (1992). We chose them as the most promised approaches: Czyż et al. (2023) argue that KSG is the preferred estimator for low-dimensional problems, neural estimators are preferred choices for high-dimensional problems if only a few dimensions are interacting, and the CCA-based estimator is the optimal choice for multivariate normal distributions.

**Reported uncertainty**   For each distribution we sampled $N' = 5,000$ data points $S = 10$ times to obtain different data sets on which the estimators were run. For each estimator and a task we then calculated the mean and the standard deviation (basing on the $S - 1$ degrees of freedom), which we reported in the table up to two decimal digits. We rounded the standard deviation upwards to two decimal digits to not underestimate it.

### C.2.3    Benchmark results

We present the obtained results in Table 1. Results for the BMM estimator on low-dimensional tasks are shown in Table 2.

**Problems not solved**   Interestingly, the Waves task is not solved by any estimator, with the best-performing one being KSG and underestimating the MI by 50%. Similarly, a high-dimensional Concentric distribution

(25-dimensional variables with 5 components) is not solved by any estimator. This contrasts with using a single component (Normal (25-dim, dense) and Normal (25-dim, sparse)), for which the CCA estimator can be used. Problems involving the spiral diffeomorphism remain unsolved.

**KSG** For low-dimensional problems, the KSG estimator seems preferable, which agrees with the conclusions from Czyż et al. (2023).

**Neural estimators** In high-dimensional problems (25-dimensional or 50-dimensional variables), neural estimators generally outperform KSG. However, they typically still underestimate the ground-truth mutual information and the model-based CCA estimate may be preferable in the problems involving multivariate normal distribution.

**CCA** In the proposed benchmark it is visible that a CCA-based estimator is not competitive on majority of the problems. The concentric multivariate normal distributions, designed as adversarial examples, indeed result in CCA not being able to find any mutual information. Moreover, we see that larger numbers of inliers result in worse predictions. Interestingly, for $N = 5{,}000$ data points and high-dimensional problems the CCA-based estimator can overestimate the result, by overfitting to the noise. This supports the view that regularizing may improve the estimates (see Sec. 3.4).

## C.3   Estimator hyperparameters

Czyż et al. (2023, Appendix E.4) study the effects of hyperparameters on mutual information estimators. We decided to use the histogram-based estimator (Cellucci et al., 2005; Darbellay & Vajda, 1999) with a fixed number of 10 bins per dimension and the popular KSG estimator (Kraskov et al., 2004) with $k = 10$ neighbors. Canonical correlation analysis (Kay, 1992; Brillinger, 2004) does not have any hyperparameters. Finally, we variational estimators with the neural critic being a ReLU network of variant M (with 16 and 8 hidden neurons), as it obtained competitive performance in the benchmark of Czyż et al. (2023, Appendix E.4). As a preprocessing strategy, we followed Czyż et al. (2023, Appendix E.3) and transformed all samples to have zero empirical mean and unit variance along each dimension. Our code is based on the MIT-licensed code associated with the Czyż et al. (2023) publication.

## C.4   Variational estimators of mutual information

In Sec. 3.3 we study the loss of Belghazi et al. (2018) as well as two other loss functions. This section provides a more detailed overview of these approaches. At the same time, our review is not exhaustive: many more variational lower bounds exist and have been described in the literature. For a detailed overview we refer to the articles by Poole et al. (2019) and Song & Ermon (2020), as well as to Chapters 4 and 7 of the textbook by Polyanskiy & Wu (2022).

Belghazi et al. (2018) use the Donsker–Varadhan loss,

$$I_{\mathrm{DV}}(f) = \mathbb{E}_{P_{XY}}[f] - \log \mathbb{E}_{P_X \otimes P_Y}\left[\exp f\right],$$

which is a lower bound on $\mathbf{I}(X; Y)$ for any bounded function $f$. Hence, they decide to use a neural critic, i.e., a neural network $f \colon \mathcal{X} \times \mathcal{Y} \to \mathbb{R}$ maximizing the functional form to obtain an (approximate) lower bound on mutual information.

As Poole et al. (2019) describe, this bound becomes tight if $c$ is any real number and $f = \mathrm{PMI}_{XY} + c$, i.e., $I_{\mathrm{DV}}(f) = \mathbf{I}(X; Y)$, so one can approach MI estimation to optimisation $I_{\mathrm{DV}}$ over a flexible family of functions $f$ parameterized by neural networks. However, we note that as only a finite sample is available, the expectation value cannot be calculated exactly, so that any provided estimate does not need to be a lower bound on MI. Moreover, if no split into training and test set is used, then $f$ may overfit and provide biased estimates (McAllester & Stratos, 2020).

Another lower bound, introduced in Nguyen et al. (2007),

$$I_{\mathrm{NWJ}}(f) = \mathbb{E}_{P_{XY}}[f] - \mathbb{E}_{P_X \otimes P_Y}\left[\exp\left(f - 1\right)\right],$$

becomes tight for $f = \mathrm{PMI}_{XY} + 1$.

Oord et al. (2018) propose a variational approximation loss which uses a batch of $(x_i, y_i)_{i=1,\ldots,n}$ samples from $P_{XY}$ to estimate

$$I_{\text{NCE}}(f) = \mathbb{E}\left[\frac{1}{n}\sum_{i=1}^{n} \log \frac{\exp f(x_i, y_i)}{\frac{1}{n}\sum_{j=1}^{n}\exp f(x_i, y_j)}\right]$$

Here, if $f(x, y) = \text{PMI}_{XY}(x, y) + c(x)$, where $c$ is any function, then $I_{\text{NCE}}(f) \to \mathbf{I}(X; Y)$ as $n \to \infty$.

### C.5 Bayesian estimation of Gaussian mixture models

In this section we provide additional details on using BMMs to estimate mutual information and the pointwise mutual information profile.

#### C.5.1 Model description

Recall from Sec. 3.4 that Bayesian estimation of mutual information consists of the following steps:

1. Propose a parametric generative model of the data, $P_\theta := P(X, Y \mid \theta)$, and assume a prior $P(\theta)$ on the parameter space.

2. Use a Markov chain Monte Carlo method to obtain a sample $\theta^{(1)}, \ldots, \theta^{(m)}$ from the posterior $P(\theta \mid X_1, Y_1, \ldots, X_N, Y_N)$.

3. Estimate mutual information (and the PMI profile) for each $\theta^{(m)}$ using the Monte Carlo method described in Sec. 2.

4. Validate the findings using e.g., posterior predictive checks and cross-validation.

We consider the following sparse Gaussian mixture model with $K = 10$ components:

$$\begin{aligned}
\pi &\sim \text{Dirichlet}(K; 1/K, 1/K, \ldots, 1/K), \\
Z_n \mid \pi &\sim \text{Categorical}(\pi), & n &= 1, \ldots, N, \\
\mu_k &\sim \mathcal{N}\left(0, 3^2 I_D\right), & k &= 1, \ldots, K, \\
\Sigma_k &\sim \text{ScaledLKJ}(1, 1), & k &= 1, \ldots, K, \\
(X_n, Y_n) \mid Z_n, \{\mu_k, \Sigma_k\} &\sim \mathcal{N}(\mu_{Z_n}, \Sigma_{Z_n}), & n &= 1, \ldots, N.
\end{aligned}$$

Sampling a single covariance matrix $\Sigma$ from $\text{ScaledLKJ}(\sigma, \eta)$ distribution corresponds to sampling the correlation matrix $R$ from the Lewandowski-Kurowicka-Joe (LKJ) distribution (Lewandowski et al., 2009):

$$p(R) \propto (\det R)^{\eta-1},$$

sampling the scale parameters

$$\lambda_1, \lambda_2, \ldots, \lambda_D \sim \text{HalfCauchy}(\text{scale}=\sigma),$$

and then constructing the covariance matrix as $\Sigma_{ij} = R_{ij}\lambda_i\lambda_j$.

The sparse Dirichlet prior is a finite-dimensional alternative to the Dirichlet process, which truncates the number of occupied clusters depending on the data (Frühwirth-Schnatter & Malsiner-Walli, 2019). In particular, the *a priori* expected number of clusters depends on the number of data points to be observed. We generally use NumPyro (Phan et al., 2019) with local latent variables $Z_n$ marginalized out, what allowed us to run Markov chain Monte Carlo inference using the NUTS sampler (Hoffman & Gelman, 2014).

The $m$th sample is therefore given by

$$\theta^{(m)} = \left(\pi^{(m)}, \left(\mu_k^{(m)}, \Sigma_k^{(m)}\right)_{k=1,\ldots,K}\right)$$

which is then used to parametrize a Gaussian mixture distribution $P_{\theta^{(m)}}$. Finally, using the Monte Carlo method described in Sec. 2, we then estimated mutual information $\mathbf{I}(P_{\theta^{(m)}})$ and the PMI profile.

Table 2: BMM-provided estimates for selected low-dimensional problems. Each run corresponds to estimation on a different sample and reports the mean together with a credibility interval obtained from the 10th and 90th percentile. Bold font represents problems solved by BMMs.

| | True MI | Run 1 | Run 2 | Run 3 |
|---|---|---|---|---|
| **Additive** | 0.33 | 0.31 (0.29–0.34) | 0.29 (0.26–0.31) | 0.31 (0.29–0.34) |
| **X** | 0.41 | 0.42 (0.38–0.45) | 0.41 (0.38–0.45) | 0.41 (0.38–0.44) |
| **AI** | 0.78 | 0.78 (0.74–0.82) | 0.76 (0.73–0.80) | 0.78 (0.75–0.82) |
| Galaxy | 0.49 | 0.27 (0.24–0.29) | 0.29 (0.26–0.31) | 0.30 (0.27–0.32) |
| **Concentric (3-dim, 5)** | 0.46 | 0.46 (0.43–0.48) | 0.46 (0.43–0.49) | 0.45 (0.42–0.48) |
| **Concentric (5-dim, 5)** | 0.63 | 0.64 (0.61–0.68) | 0.63 (0.60–0.66) | 0.63 (0.60–0.66) |
| **Concentric (3-dim, 10)** | 0.56 | 0.56 (0.53–0.59) | 0.57 (0.53–0.60) | 0.46 (0.43–0.49) |
| Concentric (5-dim, 10) | 0.75 | 0.63 (0.60–0.66) | 0.62 (0.58–0.65) | 0.56 (0.53–0.59) |
| **Inliers (5-dim, 0.2)** | 0.63 | 0.69 (0.63–0.74) | 0.66 (0.61–0.71) | 0.64 (0.59–0.69) |
| Inliers (5-dim, 0.5) | 0.27 | 0.18 (0.15–0.21) | 0.29 (0.26–0.33) | 0.19 (0.16–0.21) |
| **Normal (5-dim, dense)** | 0.59 | 0.58 (0.54–0.62) | 0.61 (0.57–0.65) | 0.59 (0.56–0.63) |
| **Normal (5-dim, sparse)** | 1.02 | 1.02 (0.97–1.07) | 1.04 (0.99–1.09) | 1.03 (0.98–1.08) |
| **Student (1-dim)** | 0.22 | 0.24 (0.21–0.27) | 0.23 (0.20–0.26) | 0.22 (0.19–0.25) |
| **Student (2-dim)** | 0.43 | 0.43 (0.39–0.47) | 0.41 (0.37–0.44) | 0.40 (0.37–0.44) |
| Student (3-dim) | 0.29 | 0.19 (0.16–0.22) | 0.20 (0.17–0.23) | 0.22 (0.19–0.25) |
| Student (5-dim) | 0.45 | 0.26 (0.22–0.31) | 0.22 (0.19–0.25) | 0.26 (0.22–0.30) |

### C.5.2 BMMs performance on the proposed benchmark

We evaluated the performance of the BMM described above on a subset of distributions proposed in Appendix C.2. We excluded the high-dimensional distributions due to the high computational cost of fitting Markov chains: in our preliminary experiments used to select sampling hyperparameters, the chains often failed to sample properly the posterior distribution. For lower-dimensional problems we did not notice convergence problems in our preliminary runs (apart from the label-switching issues, which do not affect the predictive distribution). To reduce the computational cost, we decided to use three (rather than ten) samples from each distribution. Then, for each sample we ran a single Markov chain with 1,000 warm-up steps and 1,000 collected samples. As we ran a single chain and the model is not identifiable due to the label switching of $Z_n$ variables, we could not employ metrics diagonosing convergence failures, such as $\hat{R}$ (Vehtari et al., 2021). We then used 1,000 Monte Carlo samples to estimate the MI in each distribution.

In Table 2 we report the MI estimates across three runs. Each estimate is a summary of the posterior distribution, specifying the mean and the 10th and 90th percentile of the distribution.

Overall, we see that BMMs offer good performance and appropriate uncertainty quantification in low-dimensional problems, such as X, AI, normal and low-dimensional concentric distributions. Interestingly, even when the model is slightly misspecified (additive and low-dimensional Student distributions with tail-shortening transformation applied), we obtain good performance.

It is however important to note that the proposed approach is not universal: for a 5-dimensional concentric distribution and 10 components and 5-dimensional problem with 50% of inliers, the model, even though it is well-specified, does not always estimate the mutual information appropriately, underestimating the mutual information. We suppose that posterior predictive checking and sensitivity analysis can be used to diagnose problems with this model. We investigate this issue in more detail in Appendix C.5.3.

### C.5.3 Posterior predictive checking and estimation of pointwise mutual information profiles

As discussed in Section 3.4, BMMs allow one to do model-based estimation of both MI and the PMI profile. At the same time, they require domain expertise to propose a generative model as well as careful model criticism. In this section we further investigate these aspects, by applying the model to the X, AI, Waves and Galaxy distributions, changing the number of data points $N \in \{125, 250, 500, 1000\}$.

For each distribution we sampled $N$ data points once, ran a single Markov chain with 2000 warm-up steps and collected 800 samples. Then, for each sample we estimated MI and the profile via the Monte Carlo approximation using 100,000 samples. We visualise the observed sample, a single posterior predictive sample and posterior on mutual information and the PMI profile in Fig. 10, Fig. 11, Fig. 12 and Fig. 13.

Although the model performance is good for the X and AI distributions, which additionally suports the results obtained in Appendix C.5.3, we see that model misspecification results in unreliable estimates for the Waves and Galaxy distributions.

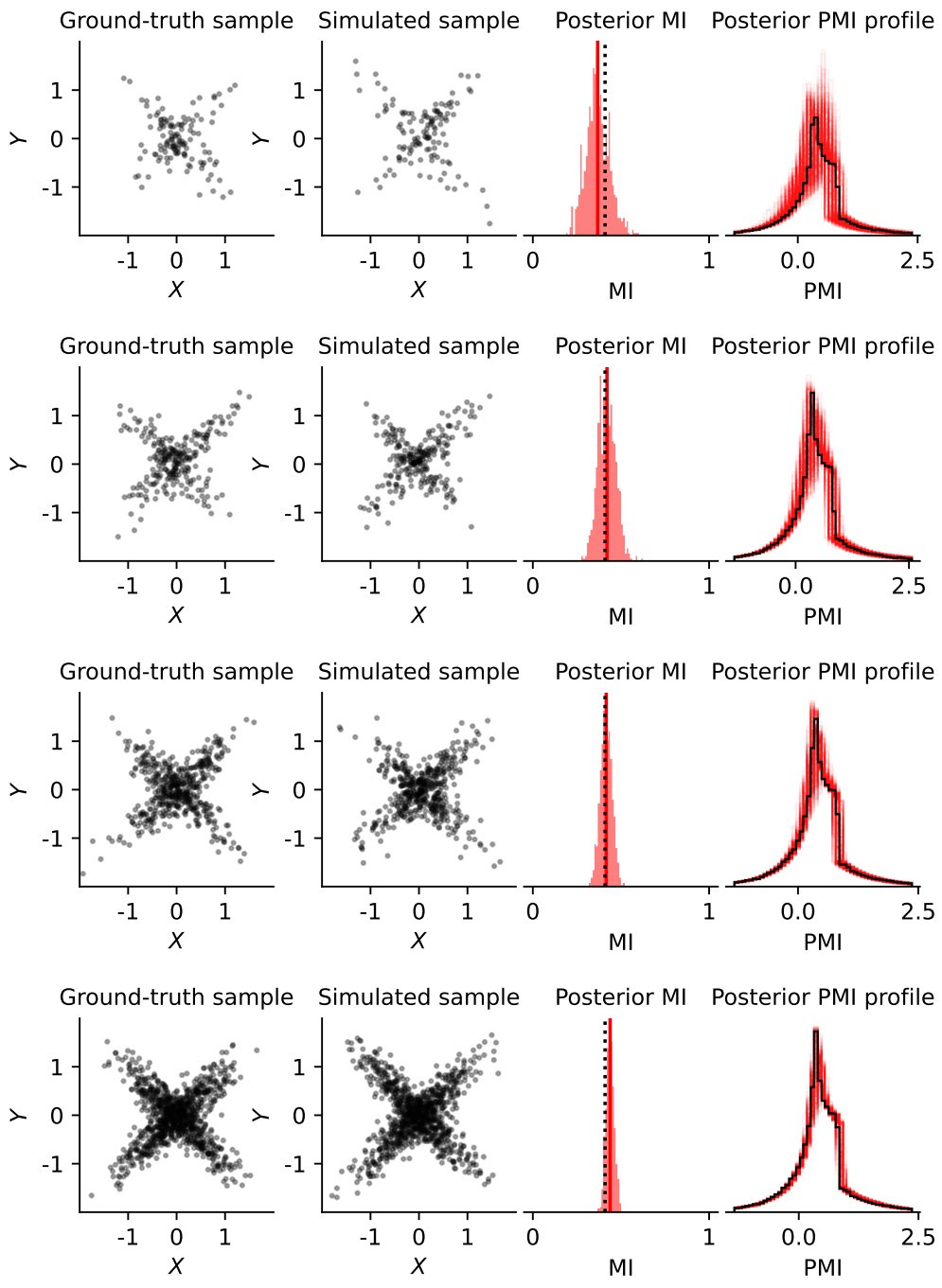

Figure 10: Gaussian mixture model fitted to the X distribution with 125, 250, 500 and 1000 samples.

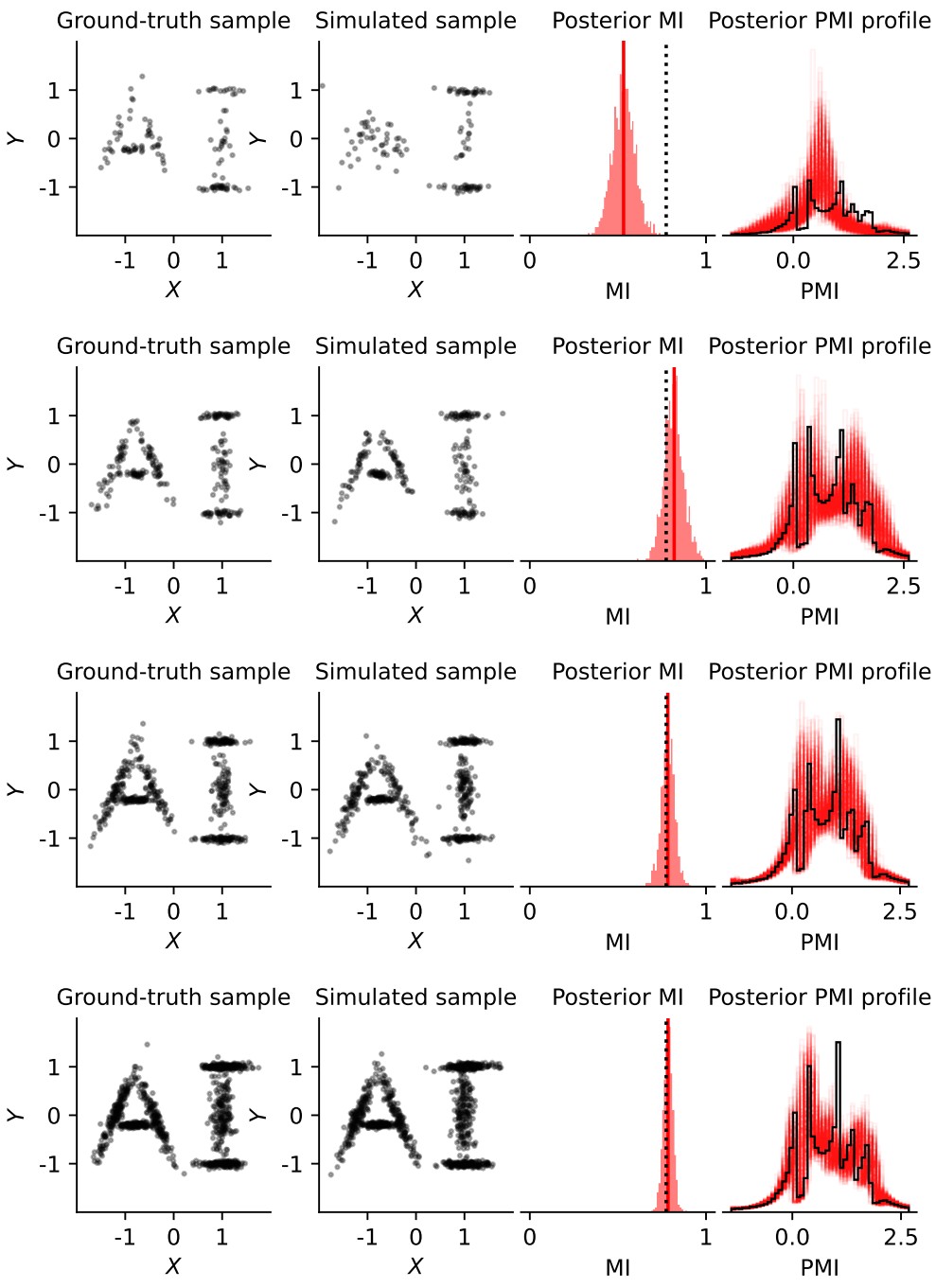

Figure 11: Gaussian mixture model fitted to the AI distribution with 125, 250, 500 and 1000 samples.

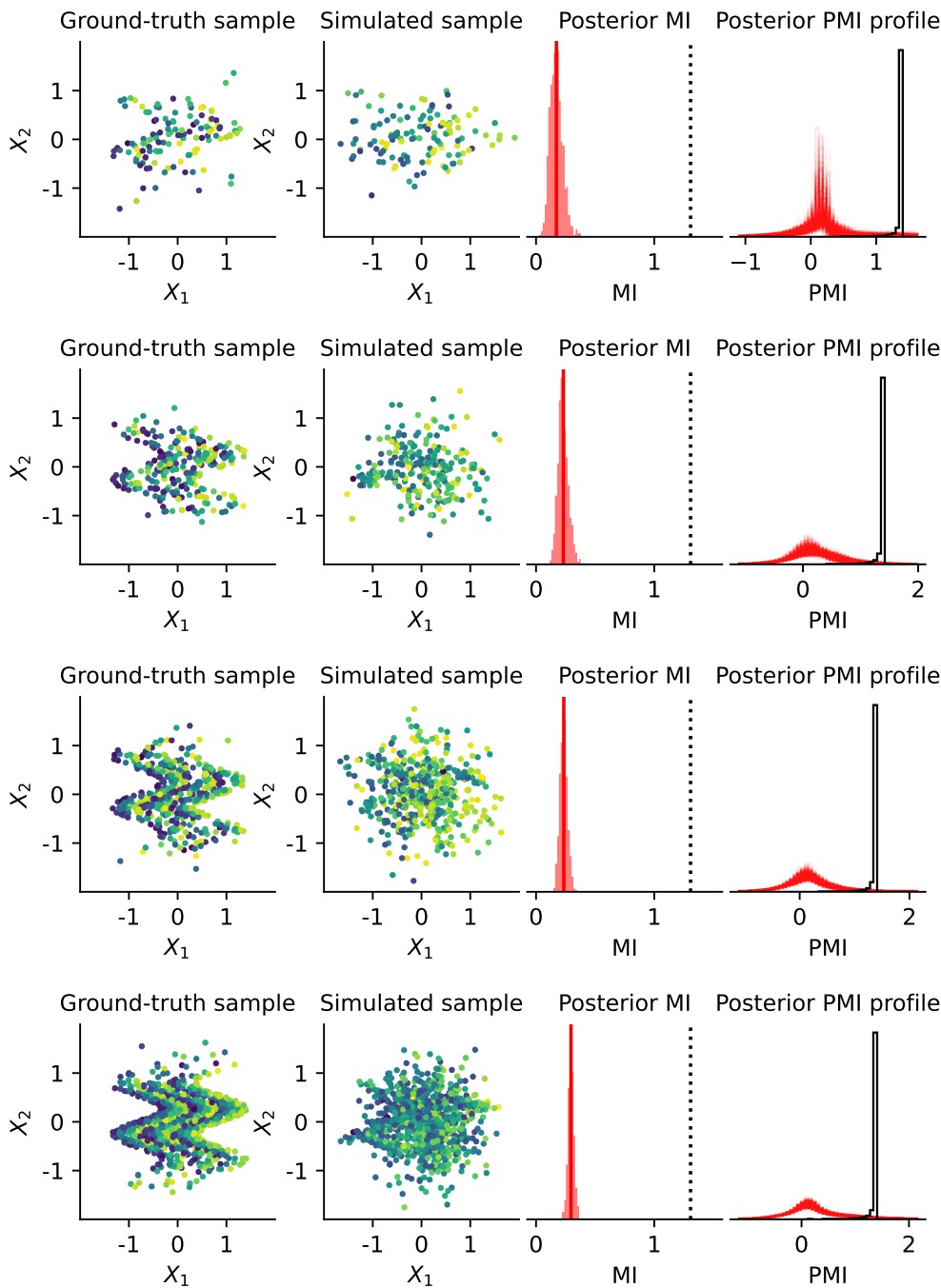

Figure 12: Gaussian mixture model fitted to the Waves distribution with 125, 250, 500 and 1000 samples.

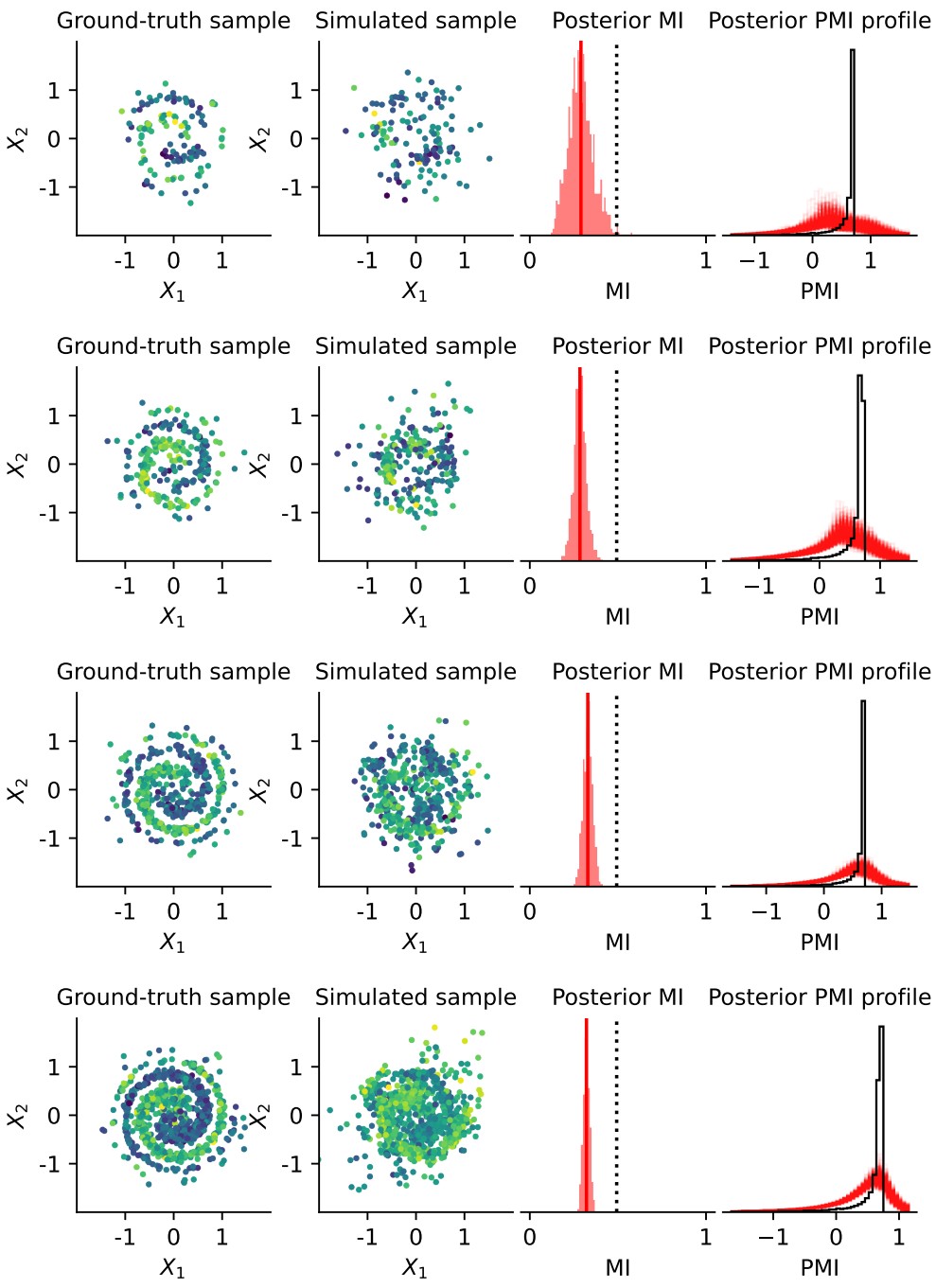

Figure 13: Gaussian mixture model fitted to the Galaxy distribution with 125, 250, 500 and 1000 samples.

