# OpenReview forum: "On the Properties and Estimation of Pointwise Mutual Information Profiles"
_TMLR — Accepted by TMLR_

### Review · Reviewer_oG5L · 2024-10-09

**Summary Of Contributions:**

The paper consists of two major parts, a characterisation of the pointwise mutual information (PMI) profile, and the definition and evaluation of a new family of models called *Bend and Mix Models* (BMMs).

For the PMI profile, the authors characterise it in greater detail than previously. Concretely, the authors establish that the profile is invariant under diffeomorphisms (thus generalising the invariance of the mutual information (MI)), and they further derive and characterise a closed-form expression for the PMI profile of jointly multivariate normal random variables.

The authors then introduce the BMMs, which combine transformations with diffeomorphisms and mixing to create new models for which the PMI profile can be efficiently estimated with Monte Carlo approximations.

Finally, the usefulness of the BMMs is illustrated in four different scenarios: as benchmarks of MI estimators, as a way to assess the robustness of MI estimators to outliers and inliers, to assess the biases and sample efficiencies of neural critics, and to provide Bayesian estimates of both the MI and the PMI profile.

**Audience:**

Yes

**Broader Impact Concerns:**

I do not have any concerns about the ethical implications of the work.

**Claims And Evidence:**

Yes

**Requested Changes:**

I would like to see a greater connection between the two parts of the paper (the PMI profile and the BMMs), or at least a justification of why they both have to be in the same paper.

Specifically for the profile, importance-wise, it seems mostly a footnote in the paper despite being mentioned as one of the key contributions. I think this is a shame because it seems to be a powerful quantity and your characterisation important. I suggest updating the paper to clarify when and where the profile is particularly useful along with some practical examples where MI estimates aren't enough. Otherwise, I suggest toning down its significance, possibly moving it to the appendix entirely or, preferably, to its own paper. Similarly, the invariance result is really interesting, but not used for anything, as far as I can see.

For the BMMs, I think you should consider dropping the name and focusing instead on extending the benchmarks of Czyż et al. (2023) with your new distributions. I also think you should extend the analysis to include additional estimators, at least those of Czyż et al. (2023), and possible additional distributions too. Otherwise, the experiments you conduct (e.g., testing the estimators' robustness to misspecification as the dataset size grows, and how well they work in higher dimensions) are very interesting and great additions to those of Czyż et al. (2023).

**Strengths And Weaknesses:**

**Strengths**
1. The paper introduces important work, which could become significant for future benchmarking of MI estimators.
2. The paper is generally very well written, and the figures are clear, understandable, and informative.
3. The authors provide a useful implementation (and promise to make it open-source).


**Weaknesses**
1. The paper consists of two parts, which appear largely disconnected to me: 1) the characterisation of the PMI profile, and 2) the introduction and evaluation of BMMs. Furthermore, each part seems too limited in scope at this point.
	1. Regarding the PMI profile, I don't fully understand why it is included in the paper. The characterisation is definitely interesting, and it seems obvious that there's a lot more information in the full profile than in the average (i.e., the MI), but its usefulness is never addressed. The profile is briefly mentioned in the experimental evaluation (Section 3), showing how well various methods can reconstruct it, but why is it important to reconstruct it at all? In which situations should we prefer the profile over its average?
	2. Regarding the BMMs, the idea is certainly interesting, but it also seems like a minor extension of Czyż et al. (2023), since this work already constructs a full benchmark based on diffeomorphisms. Extending this with mixtures is definitely interesting, but I would then expect a much more thorough evaluation, at least as comprehensive as the one in Czyż et al. (2023). The BMMs are also primarily used to estimate the MI, so I don't see the connection to the first part of the paper (the PMI profile).
2. Defining an entirely new family of distributions (the BMMs) seems unnecessary and comes across as somewhat inflating the contributions. Their sole purpose is to define distributions that can easily be sampled from such that the MI can be estimated, and mixture models and normalising flows are perhaps the most obvious ways of creating such distributions. Applying these models to estimate the ground truth MI is interesting, but I don't see why it wouldn't be sufficient to simply propose mixtures and normalising flows as two examples of models that can be used for true MI estimation.
3. As a smaller weakness, I think the paper could use a bit more comprehensive background (Section 2), for instance the basic definition of MI and why it is difficult to estimate.


*References:*
- Czyż et al., *Beyond Normal: On the Evaluation of Mutual Information Estimators*, NeurIPS 2023.

---

> ### Author Response · Authors · 2024-12-05
> **Author response**
>
> Thank you very much for the suggestions.
>
> Regarding the connection between the PMI profile and BMMs, we think about it in the following manner: the PMI profile shows that one cannot apply diffeomorphism to normal distributions, as proposed by Czyz et al. (2024), to obtain an arbitrary distribution. Hence, a wider family of models is needed (BMMs). Although BMMs do not have analytically known ground-truth MI, they do offer a convenient Monte Carlo approximation of the PMI profile, from which the MI can be extracted. As such, BMMs address the expressivity problems we found when studying the PMI profile and vice versa: the (Monte Carlo approximation of the) PMI profile is the main technical tool allowing BMMs to work. However, we agree that this connection was not clear in the submitted version and we have adjusted Sections 1 and 2 to make the connection clearer. These changes also aim at providing a bit more comprehensive background on mutual information and our aims, as we give now explicit examples of distributions for which the existing methods were not sufficient.
>
> Regarding the novelty of BMMs, we have toned down the novelty claims, instead using the name as a convenient reference for models studied in this work. Their main property is tractability of the log-density functions, which allows one to evaluate the PMI at every point. These properties could, in principle, be further relaxed, as we now detail in Section 4. We think about this property as the defining feature of a BMM, with finite mixtures and normalizing flows being two known examples. However, we anticipate that more examples of such distributions will appear in the future. Moreover, existing BMMs can be transformed and combined into larger BMMs. We think compositionality is an important thing: using only normalizing flows to estimate mutual information does not allow one to model distributions with different PMI profiles (Fig. 1) and using mixtures solely can result in misspecification (Fig. 7). Given these arguments, we do not think we could drop the name. In fact, we have already tried to not give BMMs a name in previous iterations of our manuscript and noticed that the manuscript was more verbose and less readable.
>
> Overall, we are thankful for the suggestions and we wonder what you think about the introduced changes.

---

> ### Comment · Reviewer_oG5L · 2024-12-30
>
> Dear authors,
>
> I apologise for not responding before now.
>
> Thank you very much for your updated manuscript and for addressing my concerns. Your comments and updates cleared up some confusion on my side, so I am happy to recommend acceptance.

---

> > ### Comment · Action_Editor_poZF · 2024-12-30
> > **Thank you for your follow-up, but...**
> >
> > Hi Reviewer oG5L, can you please put in a formal recommendation (it should be a button for you at the top of the page for this paper. Alternatively, you should be able to see it in the "Task" menu located at the top of the OpenReview site.
> >
> > Thanks,
> > AE

---

### Review · Reviewer_RQdA · 2024-10-11

**Summary Of Contributions:**

This manuscript introduces the notion of Pointwise Mutual Information Profile or PMI profile, and demonstrates that, much like Mutual Information or MI, the PMI profile is invariant to diffeomorphisms. Based on this insight, the manuscript highlights the limitations of MI estimation benchmarks that rely on generating varied distributions by transforming bivariate normal distributions since the PMI profile of the transformed problem remains the same. To generate MI estimation problems (for benchmarking MI estimation schemes) with varying PMI profiles, the manuscript introduces the Bend-and-Mix Models or BMMs that are mixtures of transformed closed-form distributions. While the PMI profile and the MI cannot be obtained analytically for the BMMs, the manuscript argues that these can be tractably estimated with Monte Carlo approaches.

Given these BMMs (and their corresponding PMI profiles), the manuscript highlights various interesting case studies. The first case study highlights the varying levels of performance of various MI estimation schemes on low dimensional data with different PMI profiles, including a few cases where none of the considered MI estimators perform well even in this low dimensional setting. Building upon this, the second case study highlights how BMMs can be used to study the robustness of existing MI estimators by mixing various forms (inliers vs outliers) and proportions of noise distributions. The third case study establishes a connection between the PMI and the neural critic functions used in existing variational MI estimators, and highlights the performance of the various estimators given the ground truth PMI as the neural critic under different conditions (biased vs unbiased critic, or higher dimensional data). Finally, the authors demonstrate how a family of BMMs can be used for the Bayesian estimation of the MI and the PMI profile, thereby providing uncertainty estimates for both the estimated MI and the estimated PMI profile, highlighting conditions under which this scheme can perform well or poorly.

**Audience:**

Yes

**Broader Impact Concerns:**

The manuscript does not have a Broader Impact Statement, and I do not have any concerns regarding its ethical implications.

**Claims And Evidence:**

Yes

**Requested Changes:**

- (RC1) A (theoretical or empirical) discussion motivating the need for varied PMI profiles (as discussed in W1 and W1a) would greatly improve this paper.

- (RC2) Some discussion regarding the computational cost involved in having a benchmark distribution that requires additional step of MI and PMI profile estimation would be useful.

Both my requested changes are not critical, but will greatly improve the paper in my opinion.

**Strengths And Weaknesses:**

Strengths:

- (S1) The manuscript makes a compelling case highlighting the limitations of existing benchmarks for MI estimation utilizing PMI profiles, and provides a way of generating more varied problems for MI estimation benchmarking. Of critical importance is the ability to study the robustness of the MI estimators to noisy observations, which the BMMs can nicely model, and provide a clean way of evaluating the robustness across a spectrum of noise.

- (S2) The authors not only present the notions of PMI profile and BMMs, but also do a great job at highlighting how they can be useful both theoretically and empirically. The empirical evaluations highlight the different ways that BMMs can be used for either benchmarking MI estimators or directly for MI estimation with uncertainty estimates.

- (S3) The authors do great job at presenting the ideas, the theoretical results, and the empirical case studies. Furthermore, the Appendix is well organized (and appropriately cited in the main paper) with many of the necessary details. There are many questions that I had while reading the main text that were very clearly answered when I looked in the Appendix. The Appendix also provides a very informative additional description and discussion regarding the MI estimators utilized, and their performances on the benchmark datasets, which is very useful in my opinion.

- (S4) The authors provide a useful discussion of limitations and future work.

Weaknesses:

- (W1) While the PMI profile is an interesting concept, and BMMs can be used to generate data with varied PMI profiles (as opposed to a single PMI profile in existing MI estimation benchmarks), the connection between the structure of the PMI profile and the hardness of the MI estimation task is not made explicit. Without this connection (or at least some technical discussion), it is not clear why we need distributions with varying PMI profiles. For example, one can imagine that if the PMI profile is sub-Gaussian, the mean estimation (the mean being the MI) would be easier due to concentration to the mean than with a PMI profile which is not -- as exemplified in Fig 1, where one can imagine that MI estimation is significantly easier with the blue PMI profile regardless of the transformations (note that I do not think that this is clearly true, but just the form of technical justification/discussion that I think would motivate varying PMI profiles).

- (W1a) Furthermore, it is not clear if real data where we need to estimate MI have varying PMI profiles -- it seems plausible that they would but some form of empirical validation would make the case for varying PMI profiles stronger.

- (W2) The need for Monte Carlo methods to estimate the "true" MI and the PMI profile (up to some level of approximation) of BMMs makes their use in benchmarks a bit awkward. Given the need for such additional estimation (instead of having an analytically computable solution) brings in a computation-quality tradeoff. Depending on the dimensionality and structure of the data (the complexity of the "bending" and the number of components in the "mixture"), the true MI (or PMI profile) estimation can be quite computationally challenging, and could probably introduce error in the estimation of the "ground-truth" MI that will be used as the gold-standard to evaluate MI estimators, thereby bringing the conclusions of such evaluations into question. Beyond the guarantees of Monte Carlo estimators, there does not seem to be a good way of deciding how faithful the "ground-truth MI" is -- I think the guarantees are often additive, not multiplicative, that could require large number of sample when the ground-truth is very very close to zero.

Minor comment:

- In the example corresponding to Figure 3, I think this experiment (comparing the robustness of different MI estimators) could be conducted without use of BMMs or utilizing anything related to the PMI profile. I would think that the advantages of BMMs come into play when different diffeomorphisms would be applied to the data. Is my understanding flawed?

---

> ### Author Response · Authors · 2024-12-05
> **Author response**
>
> Thank you for many excellent suggestions, we have improved the manuscript, discussing the introduced changes below.
>
> > (W1) While the PMI profile is an interesting concept, and BMMs can be used to generate data with varied PMI profiles (as opposed to a single PMI profile in existing MI estimation benchmarks), the connection between the structure of the PMI profile and the hardness of the MI estimation task is not made explicit. Without this connection (or at least some technical discussion), it is not clear why we need distributions with varying PMI profiles. For example, one can imagine that if the PMI profile is sub-Gaussian, the mean estimation (the mean being the MI) would be easier due to concentration to the mean than with a PMI profile which is not -- as exemplified in Fig 1, where one can imagine that MI estimation is significantly easier with the blue PMI profile regardless of the transformations (note that I do not think that this is clearly true, but just the form of technical justification/discussion that I think would motivate varying PMI profiles).
>
> Thank you for this comment. One reason to explore varying PMI profiles is that real world data can come from a mixture distribution, which cannot be obtained by transforming a Gaussian. On the other hand, while the benchmark of Czyż at al. (2024) mainly focuses on transforming Gaussians, they include several tasks built from a transformed, multivariate Student distribution. These tasks posed a significant challenge to all estimators, even when the authors removed long tails with a continuous transformation. This also suggests that exploring tasks beyond the transformed-Gaussian class is important when evaluating general estimators. However, it seems that the connection between the PMI profile and the hardness of the tasks is nuanced – in Section 3.3 we saw that errors in estimating the mean of the true PMI profile were much smaller then the overall errors in estimating MI. This suggests that learning the PMI profile (that is learning the PMI function in regions with high probability and/or high PMI) is the most challenging part in our tasks.
>
> > (W1a) Furthermore, it is not clear if real data where we need to estimate MI have varying PMI profiles -- it seems plausible that they would but some form of empirical validation would make the case for varying PMI profiles stronger.
>
> As we show in Fig. 1 mixtures of Gaussians result in a different PMI profile from the multivariate normal distribution. Hence, we expect that many data sets which have an underlying discrete structure will have non-trivial PMI profiles. For example, the cited paper of Grabowski et al. (2019; Fig. 3) shows a model of an eukaryotic cell responding to an external discrete signal, which results in the mixture-of-Gaussians-like structure. Additionally, in Appendix B.1 we discuss the case of discrete-continuous distributions, referring to the particle physics applications of Carrara and Ernst (2023), where one of the variables is discrete.
>
> References:
>
> Frederic Grabowski, Paweł Czyż, Marek Kochańczyk, and Tomasz Lipniacki. Limits to the rate of information
> transmission through the MAPK pathway. Journal of The Royal Society Interface, 16(152):20180792,
> 2019. doi: 10.1098/rsif.2018.0792. URL https://royalsocietypublishing.org/doi/abs/10.1098/rsif.
> 2018.0792.
>
> Nick Carrara and Jesse Ernst. Using Monte Carlo tree search to calculate mutual information in high
> dimensions, 2023. arXiv: https://arxiv.org/abs/2309.08516

---

> > ### Author Response · Authors · 2024-12-05
> > **Author response**
> >
> > > (W2) The need for Monte Carlo methods to estimate the "true" MI and the PMI profile (up to some level of approximation) of BMMs makes their use in benchmarks a bit awkward. Given the need for such additional estimation (instead of having an analytically computable solution) brings in a computation-quality tradeoff. Depending on the dimensionality and structure of the data (the complexity of the "bending" and the number of components in the "mixture"), the true MI (or PMI profile) estimation can be quite computationally challenging, and could probably introduce error in the estimation of the "ground-truth" MI that will be used as the gold-standard to evaluate MI estimators, thereby bringing the conclusions of such evaluations into question.
> >
> > We do agree that when the ground-truth MI is close to zero, the number of Monte Carlo samples needed to obtain the relative error of e.g., 1%, may be prohibitively large. Similarly, when the number of mixture components used is large, then evaluation of the PMI value at a given point may also be too expensive to collect the required number of samples.
> >
> > However, in the problems considered in this work the MI values are large enough to obtain small relative Monte Carlo standard error (MCSE). In a way, this problem is relatively easy for a Monte Carlo estimator: independently of the dimensionality of X and Y, the PMI profile is one-dimensional, making estimation of the MI (together with the Monte Carlo standard error) efficient (provided that the model is a BMM, so we can efficiently sample from the $P_XY$ distribution; see also the response to RC2, where we discuss numerical accuracy).
> >
> > Overall, we think about BMMs in the following manner. There exist:
> > Expressive distributions (e.g., large image data sets), but for which ground-truth MI is not available.
> > Distributions (e.g., the ones proposed in the cited benchmark of Czyż et al. (2024)), which are based on transforming distributions with analytically known MI through mappings which do not change it. However, these transformations do not change the PMI profile, showing a limitation of this approach.
> >
> > Hence, BMMs aim to fill the niche in the middle: trading off the exact value of the MI for expressiveness of the distribution. As our experiments suggest, this trade-off in most cases is not critical as the accurate approximations to the ground-truth are efficient to obtain.
> >
> > > Minor comment:
> > > In the example corresponding to Figure 3, I think this experiment (comparing the robustness of different MI estimators) could be conducted without use of BMMs or utilizing anything related to the PMI profile. I would think that the advantages of BMMs come into play when different diffeomorphisms would be applied to the data. Is my understanding flawed?
> >
> > In Fig. 3 the noise is introduced through mixing a Gaussian “noise” component with the “signal” distribution. Hence, the constructed mixture does not have an analytically available MI and it was necessary to use the Monte Carlo approximation of the MI available for BMMs.
> >
> > > (RC1) A (theoretical or empirical) discussion motivating the need for varied PMI profiles (as discussed in W1 and W1a) would greatly improve this paper.
> >
> > We have added examples of distributions which have been studied before but which cannot be modelled by transforming a Gaussian distribution.
> >
> > > (RC2) Some discussion regarding the computational cost involved in having a benchmark distribution that requires additional step of MI and PMI profile estimation would be useful.
> >
> > This is a good point, we have added a brief discussion of the computational cost to Appendix C2.2. It was however minor: on a standard laptop, estimating the mutual information of the Galaxy task from Figure 2 as $0.4953 \pm 0.0012$ (relative MCSE smaller than 0.5%, using 200,000 samples) takes less than 2.5 seconds on a standard laptop CPU and estimating the mutual information of the 25-dimensional task Concentric from the benchmark in Appendix C2 as $1.1911 \pm 0.0015$ (relative MCSE less than 0.2%, using 200,000 samples) takes less than 3.5 seconds.

---

> > > ### Comment · Reviewer_RQdA · 2024-12-17
> > > **Thank you for the revision and the responses**
> > >
> > > I want to thank the authors for their responses and their revised manuscript:
> > >
> > > - The revision addresses a lot of my questions by providing appropriate discussions regarding the limitations of the Monte Carlo estimation of the MI and the PMI profile, which I find very helpful.
> > > - The additional discussion in Section C.2.2 has clearly addressed my question regarding the computational cost of computing a good estimate of the ground-truth MI. Furthermore, for problems in the benchmark, I think one would need to estimate the ground-truth MI only once, so the cost can be amortized over repeated use of the benchmark. Of course, there is an issue with handling problems where the ground-truth MI is zero, which is an interesting class of problems in and of itself.
> > > - Finally, the author response addresses my question regarding the need for benchmarks that have data with varied PMI profiles, especially given that mixtures of Gaussians (and other distributions with discrete structures) can have non-Gaussian PMI profile.
> > >
> > > Overall, I think the authors have addressed my requested changes appropriately.

---

> > > > ### Author Response · Authors · 2024-12-18
> > > >
> > > > Thank you very much for all the great suggestions and the kind words!

---

### Review · Reviewer_xnSm · 2024-11-23

**Summary Of Contributions:**

This paper is concerned with the study of Pointwise Mutual Information (PMI) between two random variables $X$ and $Y$, which generalizes Mutual Information (MI). The PMI function is defined as $PMI_{X,Y}(x, y) = \log P_{X,Y} (x, y) / \log P_X(x) \log P_Y(y)$ and the PMI profile is the distribution of $PMI_{X,Y}(X,Y)$. In particular, the expectation of the PMI profile is the MI itself.

The authors prove several properties of the PMI profile, the most important one being invariance through diffeomorphisms (bending) and mixtures. This justifies the definition of "Bend and Mix Models" (BMMs), for which MI admits a tractable, unbiased Monte-Carlo estimator.
By combining simple distributions into more sophisticated ones, BMMs yield more challenging benchmarks for MI estimation. The need for such benchmarks is supported by a series of numerical experiments, in which existing MI estimators are compared on BMM-generated data.

**Audience:**

Yes

**Broader Impact Concerns:**

No broader impact concerns.

**Claims And Evidence:**

Yes

**Requested Changes:**

## Major

1. In the whole manuscript, the claims related to novely / tractability / universality of BMMs should be toned down.

## Minor

1. The colorblind-friendliness of some figures (2-right, 3, 5) should be improved, e.g. by using different markers / linestyles in addition to colors (the present reviewer is partly colorblind).
1. In Section 2.1, Theorem 4, canonical correlations should be defined.
1. In Section 3.2, the following sentence seems incorrect: Monte-Carlo estimation is not exact.
> Alternatively, we can model the system as a BMM and evaluate this quantity *exactly*.

## Questions

1. In Section 3.1, how do you explain that no estimator can accurately represent the waves task? Is it a qualitative limitation (insufficient expressiveness), or just a quantitative one (insufficient training, waves too close together, etc.)?
1. In Section 3.2, Proposition 11, is the upper bound tight?
1. In Section 3.3, I found this remark particularly interesting. Do you have any evidence for it beyond the empirical one you present, even vague theoretical intuitions?
> This suggests that although the neural critics may not learn the PMI function properly in regions with low density, the PMI profile (and, hence, the MI) can still be approximated well.
1. In Section 3.4, can you explain or demonstrate why the plug-in estimator is biased for discrete data?
1. In Section 3.4, does your proposed methodology lead to 2 nested layers of Monte-Carlo, one for $\theta$ and one for $X, Y | \theta$?
1. In Section 4, can you clarify the following sentence?
> However, one could imagine a top-down approach, starting with a more general distribution PXY for which only unbiased estimators for the log-densities are available.

**Strengths And Weaknesses:**

**Disclaimer:** I am not familiar with the previous state of the art on MI estimation, so I cannot comment on novelty. In particular, I don't know whether the PMI profile has been studied at all before (the literature review presented by the authors seems to suggest it hasn't).

## Strengths

The paper is well written and easy to read.
Its theoretical part contains elegant and non-trivial results, such as the PMI profile of a multivariate normal. The connection with normalizing flows is also pleasant.
Its experimental part makes a compelling case in favor of non-trivial distributions for MI estimation benchmarks. Indeed, data sampled from BMMs lets us discriminate between estimation algorithms which would perform similarly on simpler tasks. Additionally, the PMI profile naturally gives rise to uncertainty estimates for MI, which other methods don't always provide.

## Weaknesses

To me, the main flaw of this article lies in the exposition, which gives center stage to the family of BMMs. I don't think that is where the true contribution is.
Suppose I have two random variables $X_1$ and $X_2$ for which I can draw samples and compute densities. The fundamental idea behind BMMs is that two simple transformations also behave nicely:

| | drawing samples | computing densities |
| --- | --- | --- |
| diffeomorphism $f(X_1)$ | always possible | possible if $\det J_f$ is tractable |
| mixture $w_1 X_1 + w_2 X_2$ | always possible | always possible |

This fact is easy to prove and has no doubt been used countless times before, especially in the literature around graphical models, probabilistic circuits, normalizing flows, etc. The same might hold for two-step Bayesian estimation of MI (first $\theta$, then $PMI | \theta$) but I'm less confident about it.

More problematically, BMMs are not _sufficient_ to make MI estimation tractable: even if the Monte-Carlo estimator is unbiased, it may require many samples to provide accurate values. This is of course mentioned by the authors, but not prominently enough. Finally, the definition of the BMM family is still rather vague, because "Jacobian tractability" is not defined univocally.

In my view, the focus should be shifted from the BMMs themselves to their experimental applications. In the case of benchmark tasks, it is fine to let the Monte-Carlo simulation run for $N = 200 000$ samples, because we need to compute a precise ground truth. What matters after that is not how we computed this ground truth, but how various estimators measure up to it. However, I am eager to discuss this point with the authors and understand their perspective.

---

> ### Author Response · Authors · 2024-12-05
> **Author response (Part 1)**
>
> Thank you very much for many insightful comments and questions. We discuss them below.
>
> **Major changes**
> > 1. In the whole manuscript, the claims related to novely / tractability / universality of BMMs should be toned down.
>
> Thank you, we have adjusted the abstract, introduction, and Section 2.2 accordingly.
>
> **Minor changes**
> > 1. The colorblind-friendliness of some figures (2-right, 3, 5) should be improved, e.g. by using different markers / linestyles in addition to colors (the present reviewer is partly colorblind).
>
> Thank you very much for this comment and we apologize for this mistake. We have changed the colour scheme and added the markers.
>
> > 2. In Section 2.1, Theorem 4, canonical correlations should be defined.
>
> We added a reference for canonical correlations prior to Theorem 4.
>
> > 3. In Section 3.2, the following sentence seems incorrect: Monte-Carlo estimation is not exact.
> Alternatively, we can model the system as a BMM and evaluate this quantity exactly.
>
> Thank you, we have corrected this sentence.
>
> **Questions**
>
> > 1. In Section 3.1, how do you explain that no estimator can accurately represent the waves task? Is it a qualitative limitation (insufficient expressiveness), or just a quantitative one (insufficient training, waves too close together, etc.)?
>
> The answer depends on the estimator, since there is a tradeoff between an estimator's generality (for which tasks it can converge given enough data) and how much data it needs to converge. Histogram and KSG are guaranteed to converge given sufficient data (which might mean prohibitively large amounts of data, especially in high dimensions). InfoNCE and MINE are guaranteed to converge given enough data and a sufficiently large neural network (which allows for sufficient expressiveness). The CCA estimator fails because it assumes that the joint distribution is Gaussian distribution (insufficient expressiveness).
>
> > 2. In Section 3.2, Proposition 11, is the upper bound tight?
>
> The bound is tight whenever we can unambiguously distinguish noise from signal. If for a sample $y \sim  P(Y')$ we can tell whether it came from the joint component of the mixture $P_{XY}(x, y)$ or the noise component $P_X(x) \otimes N(y)$, then the mutual information $I(X; Y')$ is exactly the mutual information of the joint $I(X; Y)$ multiplied by the mixture weight $1-\alpha$. If there is ambiguity, more information is lost and the bound is no longer tight.
>
> > 3. In Section 3.3, I found this remark particularly interesting. Do you have any evidence for it beyond the empirical one you present, even vague theoretical intuitions?
> > > This suggests that although the neural critics may not learn the PMI function properly in regions with low density, the PMI profile (and, hence, the MI) can still be approximated well.
>
> Thank you! We think about it in the following manner: consider a region $A\subset \mathcal X\times \mathcal Y$ such that $P_{XY}(A)$ is small. Hence, there are not many samples in this region available and the behaviour of the PMI function cannot be reliably extrapolated to this region by neural networks. However, the contribution of this region to the overall PMI profile is small, because it is weighted by the small probability $P_{XY}(A)$.
>
> > 4. In Section 3.4, can you explain or demonstrate why the plug-in estimator is biased for discrete data?
>
> While the formal argument is given by Hutter (2001, Sec. 4), we think that the following intuition is useful: if $U$ is a random variable defined over a finite alphabet with $K$ elements, then the usual maximum likelihood estimate (which is the empirical distribution) results in a bias of the entropy estimate, $\mathbb E[\hat H] - H \approx -0.5 (K-1)/N = O(K/N)$, where there are $N$ available samples (Schuermann, 2004).
> Now, if $X$ and $Y$ are discrete random variables over the alphabets with $r$ and $s$ elements, respectively, then $(X, Y)$ is a variable defined over an alphabet with $rs$ elements and the bias in the mutual information, $I(X; Y) = 2H(X; Y) - H(X) - H(Y)$, is up to $O(rs / N)$, which is not negligible for $rs \ge N$.
>
> References:
>
>   - Hutter (2001), Distribution of mutual information. Advances in Neural Information Processing Systems, volume 14, [URL](https://proceedings.neurips.cc/paper_files/paper/2001/file/fb2e203234df6dee15934e448ee88971-Paper.pdf)
>   - Schuermann (2004), Bias Analysis in Entropy Estimation, [URL](https://arxiv.org/abs/cond-mat/0403192)

---

> ### Author Response · Authors · 2024-12-05
> **Author comment (Part 2)**
>
> > 5. In Section 3.4, does your proposed methodology lead to 2 nested layers of Monte-Carlo, one for $\theta$ and one for $X,Y\mid \theta$?
>
> Yes, indeed: first, one has to obtain a sample-based approximation to $\theta$ (e.g., using Markov chain Monte Carlo). Then, for each sample in the approximation, we need to sample from $X,Y\mid \theta$, to obtain the MI value. Hence, calculating the posterior distribution is more expensive than calculating only a single value.
>
> > 6. In Section 4, can you clarify the following sentence? However, one could imagine a top-down approach, starting with a more general distribution $P_{XY}$ for which only unbiased estimators for the log-densities are available.
>
> Thank you for pointing this out, this was indeed very unclear. We have rewritten this fragment in the manuscript, filling in the mathematical details and references.

---

> ### Comment · Reviewer_xnSm · 2024-12-05
>
> Thank you for the detailed responses and the changes made to the paper. I am satisfied with its current state and recommend acceptance.

---

> > ### Author Response · Authors · 2024-12-05
> >
> > Thank you once again for your time and many great suggestions for improvements!

---

### Author Response · Authors · 2024-12-05

We would like to thank all the Reviewers for their insightful reviews and the Action Editor for a very kind reminder. We have now uploaded a version of the manuscript with changes annotated with `latexdiff` and revised the plots to improve the clarity and accessibility.

---

### Author Response · Authors · 2025-01-18
**Camera ready version**

We would like to again thank the Reviewers and the Action Editor for their valuable suggestions and insightful feedback. We have uploaded the final, deanonymized version of the manuscript. We have corrected minor typographical errors and made slight adjustments to the plot ranges in selected Appendix figures for clarity.

---

> ### Comment · Action_Editor_poZF · 2025-01-18
> **Thank you and congratulations**
>
> Thank you authors for your patience and work throughout this process. The result is a fantastic paper. I'm happy to have been involved in the process.
>
> I have verified that your camera ready version satisfies the style requirements of TMLR and as such is fully ready for publication.
>
> In addition, I wanted to thank the authors for linking to their experimental repository to facilitate reproducibility and further research in these directions. One note that I would like to make / encourage is that you update the citation bibtex for this paper on your repository's README to reflect its peer-reviewed and publication status with TMLR. The full bibtex entry can be found at the top of this page just under the list of authors.
>
> Additionally, it is here for your reference:
> ```
> @article{
> czy{\.z}2025on,
> title={On the Properties and Estimation of Pointwise Mutual Information Profiles},
> author={Pawe{\l} Czy{\.z} and Frederic Grabowski and Julia E Vogt and Niko Beerenwinkel and Alexander Marx},
> journal={Transactions on Machine Learning Research},
> issn={2835-8856},
> year={2025},
> url={https://openreview.net/forum?id=LdflD41Gn8},
> note={}
> }
> ```
>
> Again, congratulations.
>
> Best,
> Taylor

---

> > ### Author Response · Authors · 2025-01-18
> >
> > Thank you very much for professionally handling the entire review process and for your kind words! We have updated the repository as per your suggestions.

---

### Decision · Action_Editor_poZF · 2025-01-03

**Recommendation:** Accept as is

**Comment:**

The reviewers unanimously advocated for this paper's publication following discussion with the authors. The authors have produced an improved version of their paper during this discussion paper and there were no remaining concerns from the reviewers.

One comment by a reviewer that I want to elevate at this point was that the authors have made a promise to open source their implementations of PMI. I mention it here to encourage the authors to follow-through on this.

**Audience:**

One of several positive contributions pointed out by the reviewers was the introduction of a new paradigm for benchmarking Mutual Information estimation approaches. The work presented in this paper highlights important limitations of existing MI benchmarks, communicating a need to revisit how MI approaches are validated and presented to the community. The proposed MI estimation technique ("Pointwise Mutual Information Profile"; PMI) should be of interest to the community as it admits more robust estimators when they may not be an analytically known MI.

**Claims And Evidence:**

Yes, there was a consensus among reviewers that all claims made in the paper were supported by the empirical evidence provided in the paper. There was some concern about the scope of dually emphasizing PMI and the BMM approach but this was addressed and clarified by the authors. In total, the authors have produced a compelling extension of prior work and sufficiently justify it with clear empirical demonstrations.